# AKAP6 orchestrates the nuclear envelope microtubule-organizing center by linking golgi and nucleus via AKAP9

Silvia Vergarajauregui[1], Robert Becker[1], Ulrike Steffen[2], Maria Sharkova[1], Tilman Esser[1], Jana Petzold[1], Florian Billing[1], Michael S Kapiloff[3], George Schett[2], Ingo Thievessen[4,5], Felix B Engel[1,5]*

[1]Experimental Renal and Cardiovascular Research, Department of Nephropathology, Institute of Pathology, Friedrich-Alexander-Universität Erlangen-Nürnberg (FAU), Erlangen, Germany; [2]Department of Internal Medicine 3 - Rheumatology and Immunology, Friedrich-Alexander-University Erlangen-Nürnberg (FAU) and Universitätsklinikum Erlangen, Erlangen, Germany; [3]Departments of Ophthalmology and Medicine, Stanford Cardiovascular Institute, Stanford University, Palo Alto, United States; [4]Department of Physics, Friedrich-Alexander-Universität Erlangen-Nürnberg, Erlangen, Germany; [5]Muscle Research Center Erlangen (MURCE), Erlangen, Germany

**Abstract** The switch from centrosomal microtubule-organizing centers (MTOCs) to non-centrosomal MTOCs during differentiation is poorly understood. Here, we identify AKAP6 as key component of the nuclear envelope MTOC. In rat cardiomyocytes, AKAP6 anchors centrosomal proteins to the nuclear envelope through its spectrin repeats, acting as an adaptor between nesprin-1α and Pcnt or AKAP9. In addition, AKAP6 and AKAP9 form a protein platform tethering the Golgi to the nucleus. Both Golgi and nuclear envelope exhibit MTOC activity utilizing either AKAP9, or Pcnt-AKAP9, respectively. AKAP6 is also required for formation and activity of the nuclear envelope MTOC in human osteoclasts. Moreover, ectopic expression of AKAP6 in epithelial cells is sufficient to recruit endogenous centrosomal proteins. Finally, AKAP6 is required for cardiomyocyte hypertrophy and osteoclast bone resorption activity. Collectively, we decipher the MTOC at the nuclear envelope as a bi-layered structure generating two pools of microtubules with AKAP6 as a key organizer.

*For correspondence:
felix.engel@uk-erlangen.de

Competing interests: The authors declare that no competing interests exist.

## Introduction

Microtubule organization plays a crucial role in cell differentiation by regulating diverse cellular processes such as cell polarization, migration, mechanotransduction, organelle positioning, and intracellular transport (*Tillery et al., 2018*). The centrosome represents the dominant microtubule-organizing center (MTOC) in proliferating animal cells (*Bornens, 2012*). However, upon differentiation, many cell types organize their microtubules at non-centrosomal sites. Non-centrosomal MTOCs (ncMTOCs) are, for example, found in axons and dendrites of neurons (*Baas et al., 1988*; *Sánchez-Huertas et al., 2016*), around the nuclear envelope of striated muscle cells (*Becker et al., 2020*; *Tassin et al., 1985a*) and osteoclasts (*Mulari et al., 2003*), as well as at the apical surface and the Golgi of epithelial cells (*Bacallao et al., 1989*; *Chabin-Brion et al., 2001*). Although non-centrosomal microtubule organization is a hallmark of many differentiated cell types, the mechanisms regulating the switch from centrosomal MTOC to ncMTOC as well as the molecular composition and regulation of ncMTOCs are just beginning to be elucidated.

DOI: https://doi.org/10.7554/eLife.61669

During late embryonic and early postnatal mammalian development, cardiac output increases, resulting in a substantial increase in mechanical load experienced by cardiomyocytes. This is associated with cell cycle exit and their terminal differentiation (*Zimmermann, 2013*), where cardiomyocytes switch from hyperplastic to hypertrophic growth (*Li et al., 1996*). Cell cycle exit is accompanied by a fundamental reorganization of the microtubule cytoskeleton characterized by centrosome disassembly, centriole splitting, and gradual localization of the centrosomal proteins PCM1, pericentrin (Pcnt), and Cdk5Rap2 to the nuclear envelope (*Zebrowski et al., 2015*). Concomitant with centrosomal protein relocalization, the main MTOC activity is found at the nuclear envelope (*Kronebusch and Singer, 1987*; *Zebrowski et al., 2015*). The nuclear envelope of mammalian osteoclasts has also been reported to recruit Pcnt and nucleate microtubules (*Mulari et al., 2003*). The best studied cell type exhibiting a nuclear envelope MTOC is the skeletal muscle cell. AKAP9 (AKAP450), PCM1, Pcnt, Cdk5rap2, Cep170, γ-tubulin, and ninein are centrosomal proteins reported to localize to the nuclear envelope during myocyte differentiation (*Bugnard et al., 2005*; *Fant et al., 2009*; *Gimpel et al., 2017*; *Srsen et al., 2009*). Nesprin-1α, a member of the LINC (Linker of Nucleoskeleton and Cytoskeleton) complex, is required for centrosomal protein localization and microtubule nucleation at the nuclear envelope, which contributes to proper myonuclear positioning (*Espigat-Georger et al., 2016*; *Gimpel et al., 2017*). Based on siRNA-mediated knockdown experiments, Gimpel et al. concluded that AKAP9 mediates microtubule nucleation at the nuclear envelope of skeletal muscle cells (*Gimpel et al., 2017*).

Along with centrosomal proteins and MTOC activity, the Golgi is localized to the nuclear envelope into a belt-like ring structure in differentiated skeletal myocytes, cardiomyocytes, and osteoclasts (*Kronebusch and Singer, 1987*; *Mulari et al., 2003*; *Tassin et al., 1985b*). To the best of our knowledge, the presence of the Golgi in a belt-like ring around the nucleus has so far not been integrated in the analysis of nuclear envelope MTOCs. Considering the capacity of the Golgi to act itself as an ncMTOC (*Efimov et al., 2007*; *Rivero et al., 2009*; *Chabin-Brion et al., 2001*; *Zhu et al., 2015*; *Wu et al., 2016*) and the role of Golgi elements in microtubule nucleation in muscle cells (*Oddoux et al., 2013*), it appears mandatory to elucidate the contribution of the Golgi to the nuclear envelope MTOC.

Collectively, it remains elusive how MTOC formation at the nuclear envelope is initiated, how this ncMTOC is organized, whether the Golgi is part of this ncMTOC, and what cellular purpose the MTOC at the nuclear envelope has in cardiomyocytes and osteoclasts. The aim of this study was to shed light on the organization of a nuclear envelope MTOC. A candidate protein to orchestrate the assembly of the MTOC at the nuclear envelope is the muscle-specific A-kinase anchoring protein β (mAKAP β, AKAP6). AKAP6 localizes to the nuclear envelope in a nesprin-1α-dependent manner, and contains several spectrin-like repeat (SR) domains (*Kapiloff et al., 1999*; *Pare et al., 2005*), which are known to facilitate cytoskeletal protein assembly (*Djinovic-Carugo et al., 2002*). In addition, it has been shown that AKAP6 is associated with nesprin-1α together with PCM1, AKAP9, and Pcnt in differentiated myoblasts using a proximity-dependent biotin identification (BioID) method (*Gimpel et al., 2017*). Based on loss and gain of function experiments, protein-protein interaction studies, and quantitative analysis utilizing a variety of cell types, we show here that AKAP6 is a key component of nuclear envelope MTOC.

## Results

### AKAP6 expression is associated with ncMTOC formation in rat cardiomyocytes

To assess whether AKAP6 expression is associated with the establishment of the ncMTOC at the nuclear envelope, we analyzed the expression levels of *Akap6* throughout late embryonic and early postnatal rat cardiac development when cardiomyocytes reorganize their MTOC. RT-PCR analyses revealed a gradual increase in cardiac *Akap6* mRNA expression between E12 and postnatal day 10 (P10) (*Figure 1—figure supplement 1A–B*). Upregulation of *Akap6* was confirmed by analyzing previously published temporal expression data describing rat heart development from midgestation to P10 (*Figure 1—figure supplement 1C*; *Patra et al., 2011*). In addition, AKAP6 was detected by immunofluorescence analysis at the nuclear envelope of E15 and P3 rat cardiomyocytes in which ncMTOC formation has been initiated, indicated by nuclear envelope localization of the centrosomal

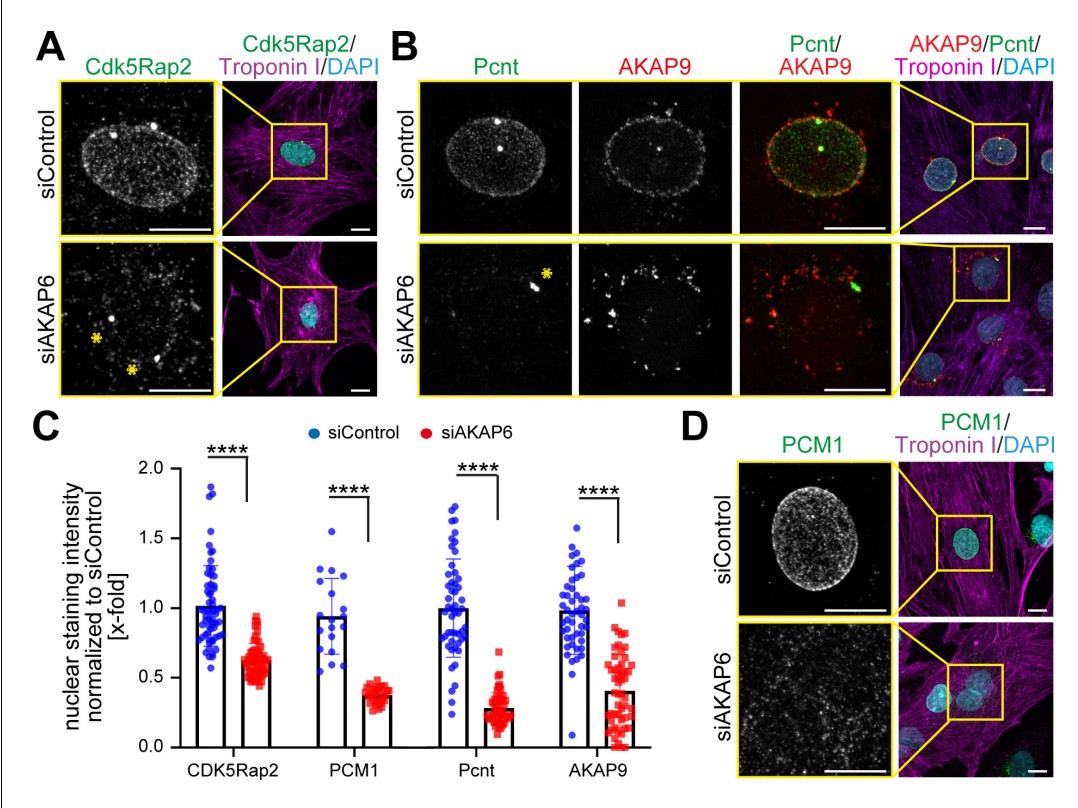

**Figure 1.** AKAP6 is required for centrosomal protein recruitment and MTOC function at the nuclear envelope. (**A–B**) Immunostaining of (**A**) Cdk5Rap2 (green) or (**B**) Pcnt (green) and AKAP9 (red) together with cardiac troponin I (magenta, cardiomyocyte-specific) and DNA (DAPI) in rat P3 cardiomyocytes transfected with control-siRNA or AKAP6-siRNA. Asterisks indicate the centrosome. (**C**) Quantification of A, B, and D as intensity of the signal at the nucleus normalized to siControl. Statistical assay: two-way ANOVA with post-hoc Bonferroni comparison. ****p<0.0001, n = 60, 60, 19, 28, 58, 46, 56, 46 (from left to right), data are pooled from three independent experiments. Error bars represent the SD. (**D**) Immunostaining of PCM1 (green), cardiac troponin I (magenta), and DNA (DAPI) in siControl- and siAKAP6-treated P3 cardiomyocytes. Scale bars: 10 μm.

The online version of this article includes the following source data and figure supplement(s) for figure 1:

**Source data 1.** Underlying data for panels D, F and G.

**Figure supplement 1.** AKAP6 expression is associated with PCM1 localization at the nuclear envelope.

**Figure supplement 1—source data 1.** Underlying data for graphs in panel 1C.

**Figure supplement 2.** AKAP6 is required for centrosomal protein recruitment and MTOC function at the nuclear envelope.

protein PCM1 (*Figure 1—figure supplement 1D*). These data support the hypothesis that AKAP6 plays a role in MTOC formation at the nuclear envelope.

## AKAP6 is required for the localization of centrosomal proteins to the nuclear envelope

In order to determine whether AKAP6 is required for the localization of centrosomal proteins to the nuclear envelope, we examined the effect of siRNA-mediated AKAP6 depletion (siAKAP6, *Figure 1—figure supplement 2A*) on several centrosomal proteins that are localized at the nuclear envelope in P3 rat cardiomyocytes (*Figure 1A–D*). Depletion of AKAP6 resulted in a marked decrease of Cdk5Rap2 and Pcnt localization at the nuclear envelope (*Figure 1A–C*), without obvious effects on their localization at the centrosome (*Figure 1A,B*, asterisks). In addition, PCM1 localization at the nuclear envelope was lost in siAKAP6-treated cardiomyocytes (*Figure 1C,D*). These data suggest that AKAP6 depletion results in the release of centrosomal proteins from the nuclear envelope into the cytosol. Consistent with this, AKAP6 depletion did not affect the total levels of PCM1 protein (*Figure 1—figure supplement 2B*).

AKAP9 was like PCM1, Pcnt, and Cdk5Rap2 lost from the nuclear envelope (*Figure 1B,C*) but redistributed to 'patchy' structures in the vicinity of the nucleus (*Figure 1B*). Nuclear envelope

localization of nesprin-1α, which is known to be required for the recruitment of centrosomal proteins to the nuclear envelope (*Espigat-Georger et al., 2016*; *Gimpel et al., 2017*), was not affected by AKAP6 depletion (*Figure 1—figure supplement 2C,D*). In order to ensure the specificity of siA-KAP6, rescue experiments were performed. Expression of an siRNA-resistant AKAP6 (AKAP6res-GFP) rescued the levels of Pcnt and AKAP9 at the nuclear envelope in siAKAP6-treated P3 cardiomyocytes comparable to those of siControl cells (*Figure 1—figure supplement 2E–H*).

Together, these data demonstrate that AKAP6 is required for the localization of centrosomal proteins to the nuclear envelope in cardiomyocytes and indicate that AKAP6 might act as an adaptor between nesprin-1α and centrosomal proteins.

## AKAP6 acts as adaptor between nesprin-1α and Pcnt or AKAP9 via its SR domains

AKAP6 contains three SR domains (SR1-3, *Figure 2A*). SR3 facilitates the recruitment of AKAP6 to the nuclear envelope by binding to nesprin-1α (AKAP6-SR3) (*Pare et al., 2005*). Expression of AKAP6-SR1-3 displaces endogenous AKAP6 from the nuclear envelope (*Kapiloff et al., 1999*; *Pare et al., 2005*). Yet, unlike siRNA-mediated AKAP6 depletion (*Figure 1B*), overexpression of AKAP6-SR1-3 in siControl-treated P3 cardiomyocytes only partially abrogated the localization of Pcnt at the nuclear envelope (*Figure 2B*, upper row). As it is known that expression of AKAP6-SR1-3 displaces endogenous AKAP6, and thus the AKAP6-bound PCNT, this result suggested that AKAP6-SR1-3 might be sufficient to anchor Pcnt to the nuclear envelope. The partially reduced localization of Pcnt at the nuclear envelope might be due to the competition of AKAP6-SR1-3 and the endogenous AKAP6 for the interaction to Pcnt. Consistent with this, expression of AKAP6-SR1-3 in AKAP6-depleted P3 cardiomyocytes rescued Pcnt and AKAP9 localization at the nuclear envelope (*Figure 2B*). These data indicate that the SR domains of AKAP6 act as an adaptor between nesprin-1α and Pcnt as well as AKAP9. Notably, AKAP9 and Pcnt are the only mammalian proteins containing the pericentrin-AKAP9 centrosome–targeting (PACT) domain (*Figure 2A*; *Gillingham and Munro, 2000*), which localizes to the nuclear envelope in cardiomyocytes when ectopically expressed (*Zebrowski et al., 2015*). Hence, we tested whether the SR domains of AKAP6 and the PACT domain of Pcnt or AKAP9 can interact. Co-immunoprecipitation experiments in HEK293 cells revealed that GFP-AKAP6-SR1-3 interacts with FLAG-Pcnt-PACT as well as FLAG-AKAP9-PACT (*Figure 2C*). Furthermore, FLAG-PACT co-immunoprecipitated with all SR1-containing GFP-AKAP6 fusion proteins (GFP-AKAP6-SR1-3, GFP-AKAP6-SR1-2, and GFP-AKAP6-SR1) but not with GFP-AKAP6-SR2 or -SR3 (*Figure 2D,E*). These data suggest that interaction between the SR1 domain of AKAP6 with the PACT domain of Pcnt and AKAP9 mediates AKAP6 anchorage of these proteins at the nuclear envelope. The direct interaction of the domains was confirmed by yeast-two-hybrid assays (*Figure 2F*). Consequently, overexpression of the SR1 domain resulted in a marked decrease of Pcnt at the nuclear envelope and the redistribution of AKAP9 to 'patchy' structures (*Figure 2G–I*), comparable to that seen in AKAP6-depleted cardiomyocytes (*Figure 1A–C*).

Collectively, these data demonstrate that AKAP6 functions as a bridge linking the PACT domain containing proteins Pcnt and AKAP9 to nesprin-1α at the nuclear envelope of cardiomyocytes.

## AKAP6 and AKAP9 form a protein platform tethering the golgi to the nuclear envelope

The Golgi itself can act as an ncMTOC (*Efimov et al., 2007*; *Rivero et al., 2009*; *Chabin-Brion et al., 2001*; *Zhu et al., 2015*; *Wu et al., 2016*). Microtubule nucleation at the cis-side of the Golgi requires AKAP9 (*Rivero et al., 2009*; *Wu et al., 2016*) and the Golgi is localized to the nuclear envelope into a belt-like ring structure in differentiated cardiomyocytes (*Kronebusch and Singer, 1987*). Therefore, we wondered whether the Golgi is part of the nuclear envelope MTOC and whether the unique 'patchy' redistribution pattern of AKAP9 in the absence of AKAP6 reflects a fragmented Golgi. Co-staining of AKAP9 and the cis-Golgi marker GM130 confirmed that AKAP9 co-localizes with Golgi fragments upon AKAP6 depletion (*Figure 3A*).

While the Golgi in siControl-treated cells was tightly connected to the nucleus in a belt-like structure, the belt-like ring was broken and widely distributed in AKAP6-depleted cells (*Figure 3A–B*). Quantification of the GM130 intensity in 0.5 μm wide concentric bands around the nuclear edge showed a distribution with a pronounced, narrow peak at the edge of the nucleus in control cells

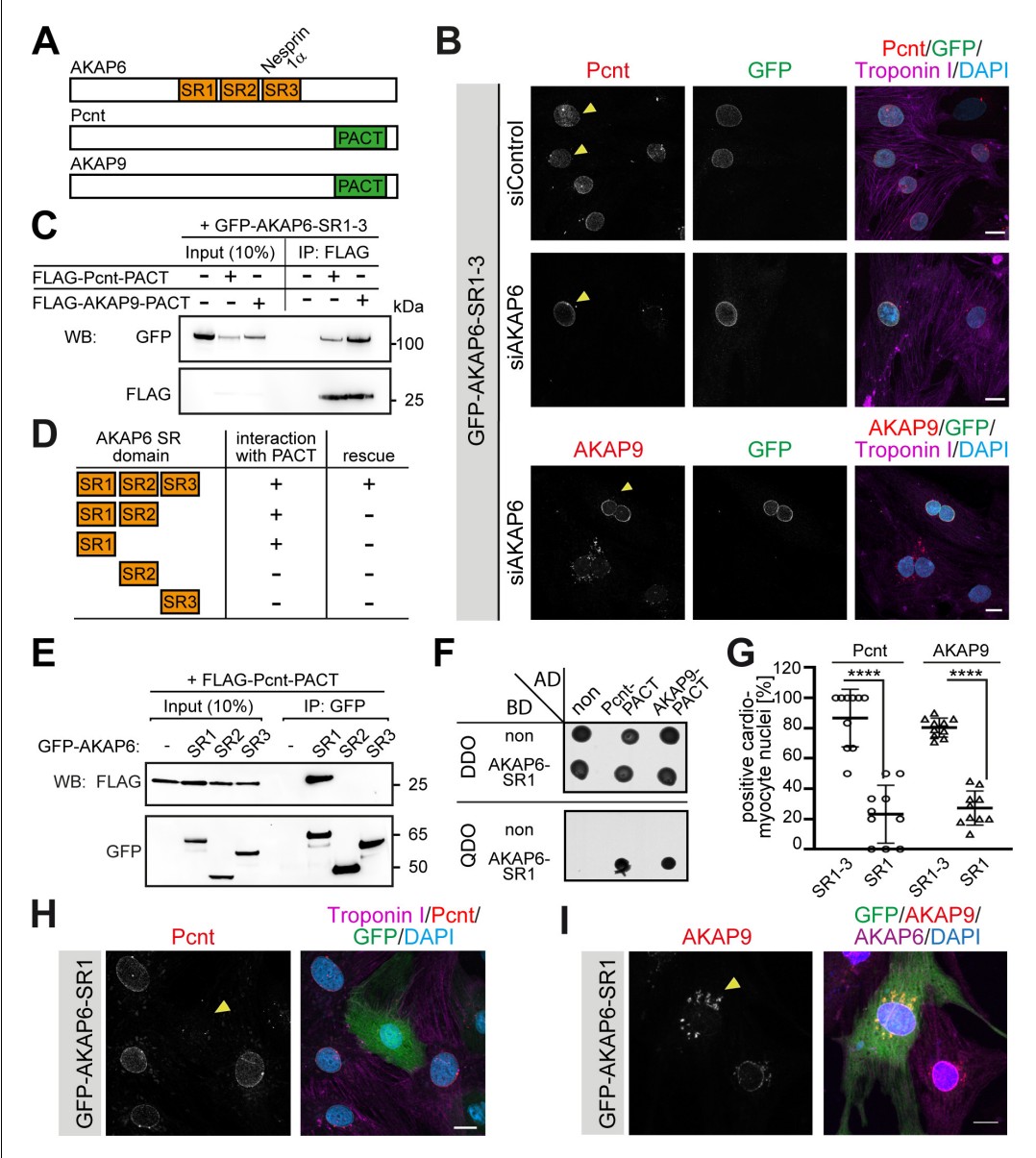

**Figure 2.** AKAP6 anchors centrosomal proteins to nesprin-1α through its SR domains. (**A**) Schematic representation of AKAP6, Pcnt, and AKAP9. (**B**) Immunostaining of Pcnt (red) or AKAP9 (red), cardiac troponin I (magenta, cardiomyocyte-specific), and DNA (DAPI) in siControl and siAKAP6-treated rat P3 cardiomyocytes transfected with GFP-AKAP6-SR1-3 suggesting that the SR domains of AKAP6 are sufficient to bind to the nuclear envelope and to anchor centrosomal proteins. Transfected cells are indicated with a yellow arrowhead. (**C**) Immunoprecipitation to demonstrate the interaction of SR1-3 of AKAP6 with the PACT domain of Pcnt or AKAP9. Lysates from HEK293 cells transfected with GFP-AKAP6-SR1-3 in the absence or presence of FLAG-Pcnt-PACT or FLAG-AKAP9-PACT were immunoprecipitated with an anti-FLAG antibody and analyzed by western blotting with antibodies against GFP and FLAG, as indicated. The experiment was performed three times (n = 3); shown is a representative image. (**D**) Schematic representation of the results in E. +: interaction; -: no interaction. (**E**) Lysates from HEK293 cells co-transfected with FLAG-Pcnt-PACT and the indicated GFP-AKAP6-SR domains were immunoprecipitated with an anti-GFP antibody and analyzed by western blotting with antibodies against FLAG and GFP; as indicated. The experiment was performed three times (n = 3); shown is a representative image. (**F**) Yeast-two-hybrid assay. Interactions were tested by monitoring the growth of yeast cells expressing AKAP6-SR1 fused to the DNA binding domain of GAL4 (BD) and Pcnt-PACT or AKAP9-PACT fused to the GAL4 activation domain (AD) proteins on DBO agar plates (double dropout; SD /-Leu /- Trp) (upper), or on QDO plates (quadruple dropout; SD /-Ade /- His/-Leu/-Trp) (lower). Growth on QDO plates indicate interaction. The experiment was performed twice (n = 2); shown is a representative image. (**G–I**) Immunostaining of (**H**) Pcnt (red) or (**I**) AKAP9, cardiac troponin I (magenta, cardiomyocyte-specific), and DNA (DAPI) in P3 cardiomyocytes transfected with GFP-AKAP6-SR1 and subsequent quantification of Pcnt- or AKAP9-positive cardiomyocyte nuclei in transfected cells (**G**) Transfected cells are labeled with a yellow arrowhead. Data are represented as individual biological replicates, together with mean ± SD. Statistical test: two-way ANOVA with post-hoc Bonferroni comparison. ****: p<0.0001. n = 10. Scale bars: 10 μm.

*Figure 2 continued on next page*

*Figure 2 continued*

The online version of this article includes the following source data for figure 2:

**Source data 1.** Underlying data for graphs in panel 2G.

(*Figure 3B*), whereas AKAP6-depleted cells showed a strongly reduced peak that was shifted towards the cytosolic side (*Figure 3B*). These data indicate that AKAP6 is required to tether the Golgi to the nuclear envelope.

The Golgi in non-muscle cells is usually organized in a ribbon that consists of clustered stacks. In mammals, formation and maintenance of the ribbon depends on microtubule-mediated processes. Treatment with reagents that depolymerize microtubules causes the redistribution of the perinuclear ribbon into dispersed ministacks (*Ayala et al., 2020*). In contrast, the belt-like ring structure of the Golgi around the nuclear envelope of skeletal myocytes is resistant to microtubule depolymerization drugs (*Tassin et al., 1985b*; *Zaal et al., 2011*), suggesting that the Golgi localization at the nuclear envelope is mediated by a microtubule-independent mechanism.

Since AKAP6 depletion resulted in the release of the Golgi from the nuclear envelope (*Figure 3A–B*) and since AKAP6 directly interacts with AKAP9 (*Figure 2C*;F), we hypothesized that AKAP6 tethers the Golgi via AKAP9 into a belt-like ring to the nuclear envelope. To test this hypothesis, we tested the effect of AKAP9 depletion on the Golgi localization and structure. AKAP9 depletion resulted in Golgi displacement and fragmentation (*Figure 3C,D*) comparable to AKAP6 depletion. In contrast, Pcnt-depleted cells showed a normal Golgi distribution (*Figure 3C,D*). In addition, overexpression of either AKAP6-SR1 or AKAP9-PACT, both interfering with the interaction between AKAP6 and AKAP9, resulted in Golgi dispersion (*Figure 3E*). In contrast, ectopic expression of the SR1-3 domain of AKAP6, which is sufficient to localize AKAP9 to the nuclear envelope, did not have any obvious effect on the Golgi (*Figure 3E*). In a converse experiment, we observed that overexpression of the Golgi-binding GM130-interacting domain of AKAP9 (GFP-AKAP9-AK1b), which is known to dissociate endogenous AKAP9 from the Golgi (*Hurtado et al., 2011*), released the Golgi from the nuclear envelope (*Figure 3F*).

Taken together, these data show that AKAP6 and AKAP9 form a protein platform tethering the Golgi to the nuclear envelope. In our model, AKAP6 binds nesprin-1α and AKAP9, which in turn, binds to GM130 (*Figure 3G*).

## Loss of AKAP6 results in the shift of the MTOC to the Golgi

The tight link of the Golgi to the nuclear envelope raises the question, whether microtubules are nucleated at the nuclear envelope, at the Golgi, or both. Analysis of microtubules in siControl-treated P3 cardiomyocytes revealed a circular, dense accumulation of microtubules around the nucleus (*Figure 4A* arrows, *Figure 4B*). In contrast, perinuclear microtubules in AKAP6-depleted cells appeared disorganized, lacking a discernible circular accumulation (*Figure 4A* arrowheads, *Figure 4B*). Quantification of the α-tubulin intensity in concentric bands around the nuclear envelope showed a reduced intensity peak at the nuclear edge and a more even distribution compared to control cells (*Figure 4B*). While these data suggested that depletion of AKAP6 alters the distribution of microtubules, it remained unclear whether nucleation of microtubules at the nuclear envelope was affected.

To address this issue and to assess the effect of AKAP6 depletion on MTOC activity at the nuclear envelope and Golgi, we performed microtubule regrowth assays after nocodazole wash-out. The main MTOC activity (α-tubulin staining) in siControl-treated cardiomyocytes was at the nuclear envelope (*Figure 4C*). Quantification analysis showed a clear α-tubulin intensity peak around the nuclear edge of control cardiomyocytes (*Figure 4D*). In contrast, in AKAP6-depleted cells, MTOC activity at the nuclear envelope was markedly reduced, α-tubulin staining was more evenly distributed towards the cytosol, and microtubules regrew mainly from AKAP9-positive Golgi patches (*Figure 4C* arrowheads, *Figure 4D*). Overexpression of the SR1 domain of AKAP6, but not the SR1-3 domain, disrupted MTOC activity at the nuclear envelope of cardiomyocytes in a similar way as depletion of AKAP6 (*Figure 4—figure supplement 1A*). In accordance with the disruption of MTOC activity, AKAP6 depletion resulted in the release of γ-tubulin from the nuclear envelope (*Figure 4E*), the primary microtubule nucleator in centrosomes (*Akhmanova and Steinmetz, 2019*), and ninein

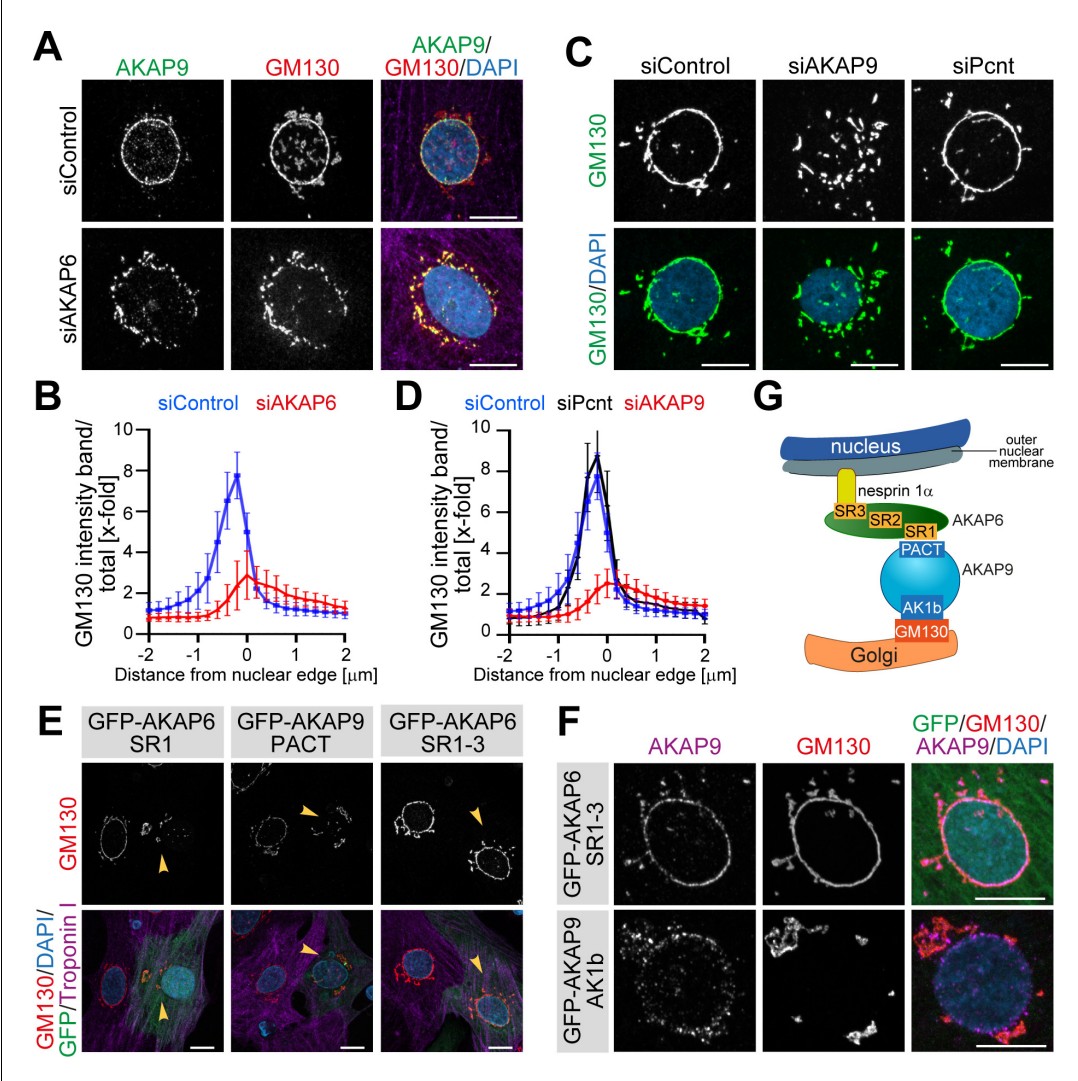

**Figure 3.** AKAP6 anchors the Golgi through AKAP9 to the nuclear envelope. (**A**) Immunostaining of AKAP9 (green), GM130 (red), and DNA (DAPI) in siControl- and siAKAP6-treated cardiomyocytes. (**B**) Quantification of A as the GM130 mean intensity in concentric bands of 0.2 μm around the nuclear edge normalized to the total GM130 mean intensity of the cell. 24 cells were analyzed per condition from three independent experiments. Error bars represent the SD. (**C**) Immunostaining of GM130 (green) and DNA (DAPI) in control, AKAP9- or Pcnt-depleted cardiomyocytes. (**D**) Quantification of C as in B. 24 (siControl), 17 (siAKAP9) and 26 (siPcnt) cells were quantified per condition and pooled from three independent experiments. Error bars represent the SD. (**E**) Immunostaining of GM130 (red), cardiac troponin I (magenta, cardiomyocyte-specific), and DNA (DAPI) in P3 cardiomyocytes transfected with GFP-AKAP6-SR1, GFP-AKAP9-PACT or GFP-AKAP6-SR1-3 as control. Transfected cells are labeled with a yellow arrowhead. (**F**) Immunostaining of GM130 (red), AKAP9 (magenta), and DNA (DAPI) in P3 cardiomyocytes transfected with GFP-AKAP6-SR1-3 or GFP-AKAP9-AK1b. (**G**) Model representing the tethering of the Golgi to the nuclear envelope through AKAP6 and AKAP9 interaction. AKAP6 bridges AKAP9 to nesprin-1α through its SR domains, while AKAP9 bridges AKAP6 to GM130 through its PACT and N-terminal (AK1b) domains. Scale bars: 10 μm.

The online version of this article includes the following source data for figure 3:

**Source data 1.** Underlying data for graphs in panels B and D.

(*Figure 4F*), a microtubule-binding protein with anchoring function (*Mogensen et al., 2000*) implicated in ncMTOC formation (*Goldspink et al., 2017*; *Ohama and Hayashi, 2009*; *Zheng et al., 2020*). Notably, ninein remained associated with AKAP9-positive patches (*Figure 4F*), which retain microtubule nucleating activity (*Figure 4—figure supplement 1B*).

Consistent with the role of AKAP9 in Golgi-dependent MTOC activity (*Rivero et al., 2009*), double depletion of AKAP6 and AKAP9 markedly inhibited microtubule outgrowth from Golgi patches (*Figure 4C*). Instead, the major MTOC activity was detected at the centrosomes (*Figure 4—figure*

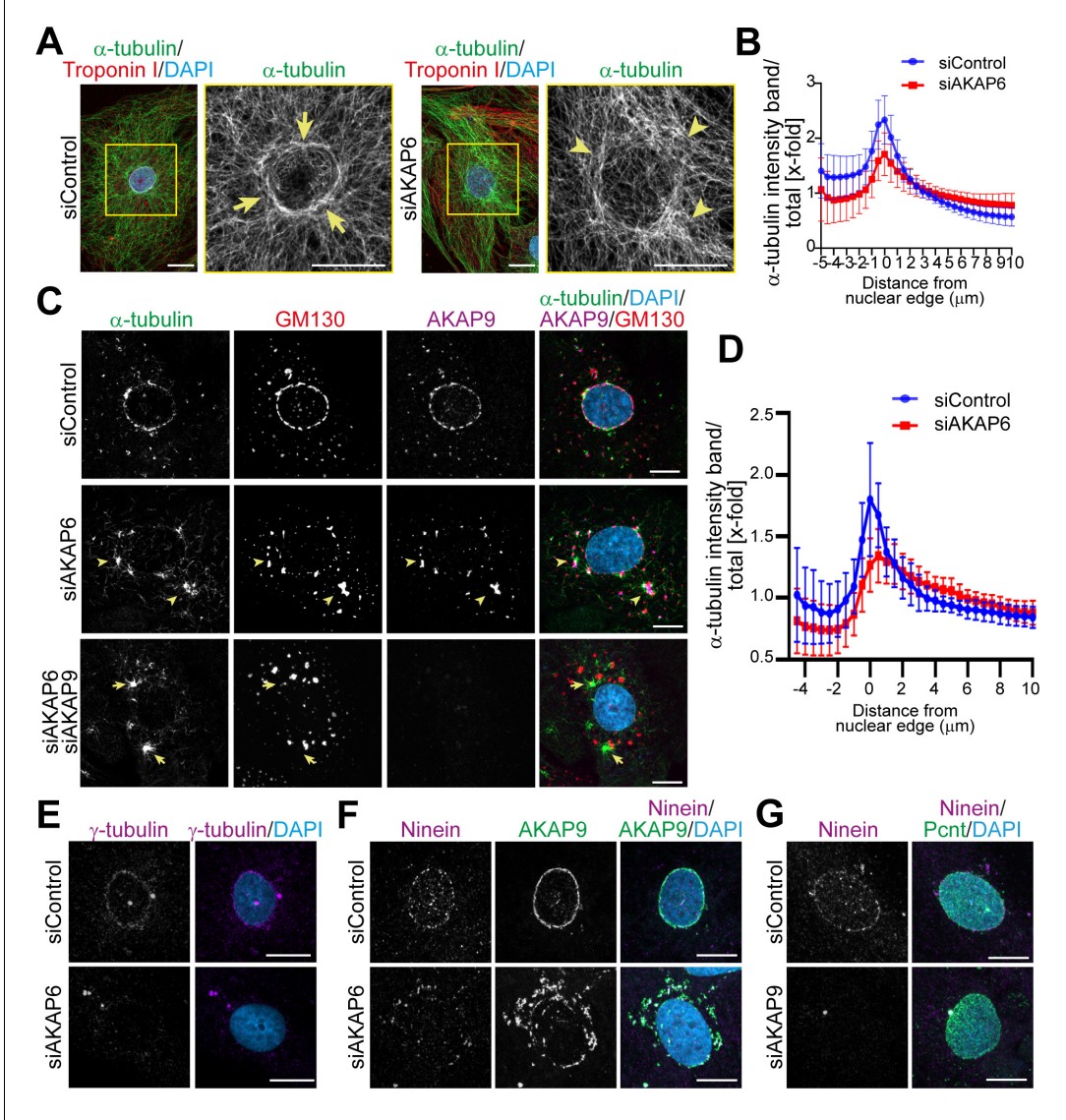

**Figure 4.** AKAP9-dependent Golgi MTOC in AKAP6-depleted cells. (A) Immunostaining of α-tubulin (green), cardiac troponin I (red, cardiomyocyte-specific), and DNA (DAPI) in siControl- or siAKAP6-treated P3 cardiomyocytes. Arrows denote the perinuclear microtubule cage in siControl cells, while arrowheads denote disorganized microtubules. (B) Quantification of A as α-tubulin intensity in concentric bands around the nucleus normalized to the total intensity of α-tubulin in the cell, 119 siControl cells and 131 siAKAP6 cells were quantified per condition from four independent experiments. Error bars represent the SD. (C) Immunostaining of α-tubulin (green), GM130 (red), AKAP9 (magenta), and DNA (DAPI) in siControl-, siAKAP6, and siAKAP6 + siAKAP9-treated P3 cardiomyocytes after 2 min of recovery from nocodazole-induced microtubule depolymerization, indicating a switch of MTOC activity from nuclear envelope to Golgi upon AKAP6 depletion (arrowheads). (D) Quantification of α-tubulin intensity in 0.5 µm wide concentric bands in AKAP6-depleted cells, as in C. 37 siControl cells and 48 siAKAP6 cells were quantified per condition from three independent experiments. Error bars represent the SD. (E) Staining of γ-tubulin (magenta) in siControl- or siAKAP6-treated cells. (F–G) Immunostaining of (F) ninein (magenta) and AKAP9 (green) in AKAP6-depleted cells or (G) ninein (magenta) and Pcnt (green) in AKAP9-depleted cells. Scale bars: 10 µm.

The online version of this article includes the following source data and figure supplement(s) for figure 4:

**Source data 1.** Underlying data for graphs in panels B and D.
**Figure supplement 1.** Enhanced centrosomal MTOC activity in AKAP6 and AKAP9 depleted cells.

supplement 1B, third row, asterisks) and at cytosolic sites that did not co-localize with GM130 (*Figure 4C*, arrows). Interestingly, AKAP9 depletion also resulted in the release of ninein from the nuclear envelope into the cytosol with no apparent organelle localization (e.g., Golgi patches) (*Figure 4G*). These data suggest that Golgi at the nuclear envelope nucleates microtubules in an

AKAP9-dependent manner and that ninein might contribute to AKAP9-dependent MTOC activity at the Golgi.

## Golgi and nuclear envelope exhibit MTOC activity utilizing different microtubule-regulating factors

Single depletion of AKAP9 did not abrogate MTOC activity at the nuclear envelope, even though it caused an evident displacement of the Golgi (*Figure 5A,B*) and an enhanced centrosomal MTOC activity (*Figure 4—figure supplement 1B*, fourth row, asterisks). This result indicates that in cardiomyocytes, the Golgi is not the only entity with MTOC activity at the nuclear envelope, and other microtubule-nucleation regulators besides AKAP9 are present at the nuclear envelope. These results differ from a previous study using skeletal muscle cells, where AKAP9 was found to be the only centrosomal protein required for microtubule nucleation at the nuclear envelope (*Gimpel et al., 2017*).

Since AKAP9 and Pcnt collaborate to bind γ-tubulin to centrosomes (*Takahashi et al., 2002*) and Pcnt is released from the nuclear envelope upon AKAP6 depletion, but not AKAP9 depletion (*Figure 4—figure supplement 1B*), we tested the contribution of Pcnt to nuclear envelope MTOC activity. Single depletion of Pcnt had no marked effect on microtubule outgrowth from the nucleus (*Figure 5A,B*). However, double depletion of AKAP9 and Pcnt markedly inhibited MTOC activity at the nuclear envelope (*Figure 5A,B*), suggesting that both proteins collaborate at the nuclear envelope for proper microtubule nucleation. To assess the relative contributions of Pcnt and AKAP9 to the nuclear envelope localization of γ-tubulin, we quantified γ-tubulin intensity at the nuclear envelope in control and Pcnt- and/or AKAP9-depleted cardiomyocytes, while monitoring microtubule outgrowth (*Figure 5C,D*). The loss of Pcnt resulted in a slight decrease of the binding of γ-tubulin to the nuclear envelope (88% compared to siControl). In comparison, loss of AKAP9 had a stronger effect on γ-tubulin binding (51%), even though microtubule nucleation was still occurring at the nuclear envelope. Interestingly, double depletion of Pcnt and AKAP9 resulted in a further decrease of γ-tubulin binding (37%), consistent with the loss of MTOC activity at the nuclear envelope (*Figure 5C,D*).

Collectively, our data demonstrate that AKAP6 regulates MTOC activity at the nuclear envelope of cardiomyocytes by recruiting AKAP9 and Pcnt, as well as linking the Golgi to the nucleus via AKAP9. Both Golgi and nuclear envelope contribute to MTOC activity at the perinuclear region of cardiomyocytes. While the Golgi-derived MTOC is dependent on AKAP9, and possibly ninein, MTOC activity at the nuclear envelope depends on the γ-TuRC-binding factors AKAP9 and Pcnt (*Figure 5E*).

## AKAP6 is a key component of nuclear envelope MTOCs

Nuclear envelope MTOCs are found in cardiomyocytes, skeletal muscle cells, and osteoclasts. AKAP6 is known as 'muscle-specific AKAP, mAKAP,' and according to the literature, AKAP6 is exclusively expressed in striated muscle cells. To study a possible role of AKAP6 as a general component of nuclear envelope MTOCs, we examined the expression of AKAP6 in osteoclasts representing a completely different cell type. Osteoclasts form by fusion of mononuclear precursors from a monocyte/macrophage lineage. Mature osteoclasts are extremely large and polarized, multinuclear cells capable of digesting calcified bone matrix (*Boyle et al., 2003*). During human osteoclast differentiation, expression of AKAP6 at the nuclear envelope was detected as early as after 3 days of differentiation in early mononucleated pre-osteoclasts with circumnuclear Golgi localization (*Figure 6A*). At 7 days of differentiation, mature multinucleated osteoclasts were present, which contained AKAP6-positive nuclei with circumnuclear Golgi localization, suggesting that AKAP6 also orchestrates the nuclear envelope MTOC in osteoclasts. Osteoclast nuclei were also positive for nesprin-1 (*Figure 6B*), consistent with the observed dependency of AKAP6 localization at the nuclear envelope on nesprin-1α in cardiomyocytes.

To test whether AKAP6 is required for nuclear envelope MTOC formation and activity in osteoclasts, we depleted AKAP6 utilizing adenoviral-mediated shRNA expression (shAKAP6). AKAP6 depletion was very efficient, and it did not affect nesprin-1 localization (*Figure 6B,C*). Loss of AKAP6 in osteoclasts resulted in the release of Pcnt from the nuclear envelope and the redistribution of AKAP9 to patchy structures in the cytoplasm (*Figure 6D,E*). Also, depletion of AKAP6 (*Figure 6F*) or overexpression of its SR1 domain (*Figure 6—figure supplement 1*) resulted in the dispersion and

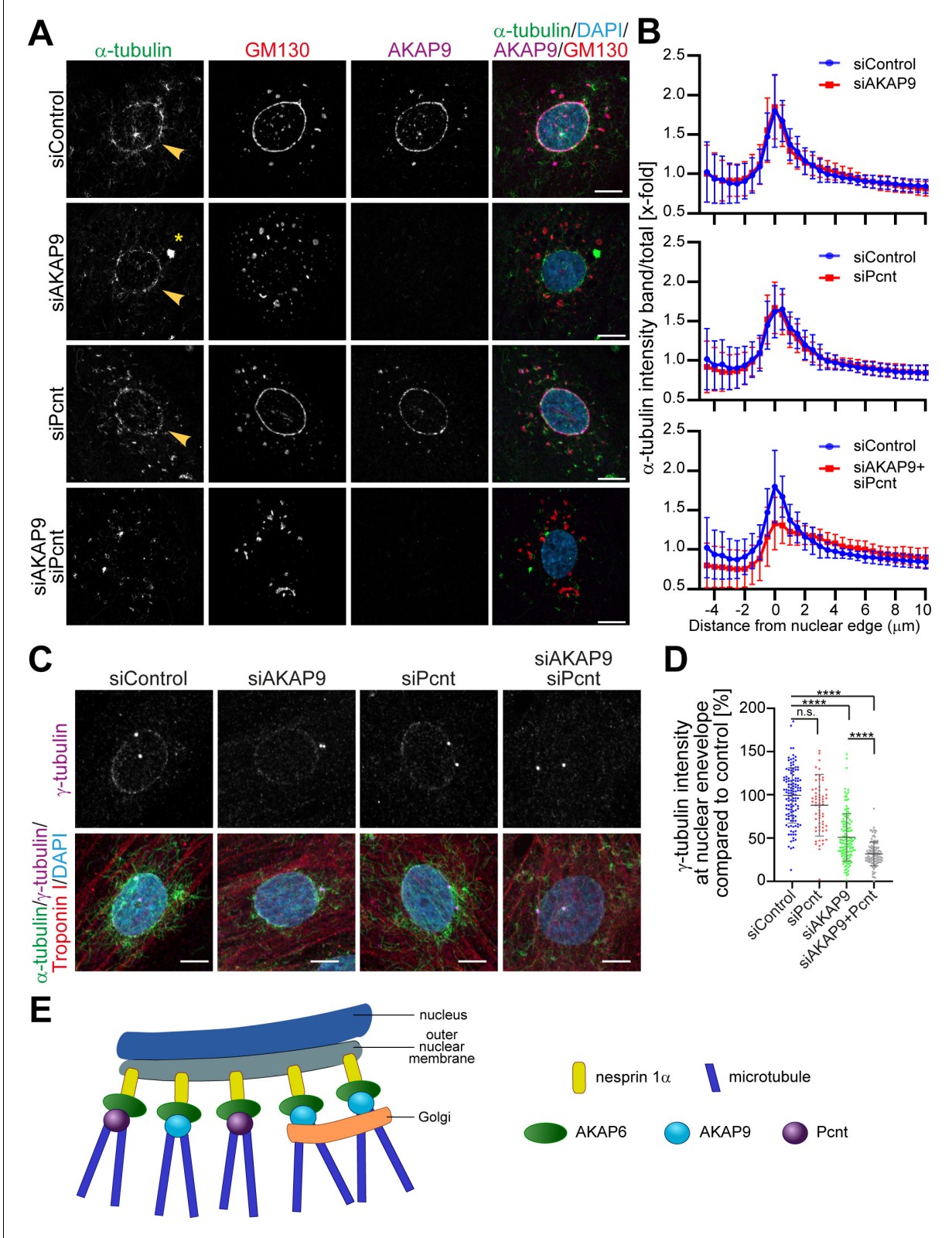

**Figure 5.** AKAP6 regulates two pools of microtubules at the nuclear envelope. (**A**) Immunostaining of α-tubulin (green), GM130 (red), AKAP9 (magenta), and DNA (DAPI) in control, as well as in the indicated siRNA-transfected cells, after 2 min of recovery from nocodazole-induced microtubule depolymerization. Asterisks indicate centrosomal MTOC and arrowheads indicate nuclear envelope MTOC. (**B**) Quantification of A as α-tubulin intensity in concentric bands around the nucleus normalized to the total intensity of α-tubulin in the cell. 37 siControl cells, 53 siAKAP9 cells, 34 siPcnt cells and

*Figure 5 continued on next page*

*Figure 5 continued*

43 siAKAP9+siPcnt cells were quantified per condition, from two independent experiments. Error bars represent the SD. (**C**) Immunostaining of γ-tubulin (magenta), α-tubulin (green), troponin I (red), and DNA (DAPI) in siRNA-treated cardiomyocytes after 2 min of recovery from nocodazole-induced microtubule depolymerization. (**D**) Quantification of C, as γ-tubulin intensity at the nuclear envelope normalized to siControl-treated cells. Statistical test: one-way ANOVA with post-hoc Bonferroni's comparison. \*\*\*\*: p<0.0001, n.s.: no significance, n = 127, 63, 136, 115 (from left to right), data are pooled from three independent experiments. Error bars represent the SD. (**E**) Model representing the two pools of microtubules regulated by AKAP6. AKAP6 orchestrates MTOC activity at the nuclear envelope by anchoring both γ-TuRC biding proteins, Pcnt and AKAP9. While Pcnt and AKAP9 cooperate at the nuclear envelope for the nucleation of microtubules, Golgi-dependent microtubule nucleation depends on AKAP9. Scale bars: 10 μm. The online version of this article includes the following source data for figure 5:

**Source data 1.** Underlying data for graphs in panels B and D.

fragmentation of the Golgi. Microtubule regrowth assays demonstrated that the main MTOC activity in osteoclasts, determined based on localization of α-tubulin and EB1-comets, is localized at the nuclear envelope (*Figure 6F*; *Mulari et al., 2003*). After AKAP6 depletion, MTOC activity at the nuclear envelope was markedly reduced and localized to Golgi patches (*Figure 6F*). Taken together, these data demonstrate that AKAP6 is also required for the formation and activity of the nuclear envelope MTOC in osteoclasts, indicating that the key role of AKAP6 in establishing nuclear envelope MTOCs is not restricted to striated myocytes.

## AKAP6 is sufficient to anchor the PACT domain of Pcnt and AKAP9 in a heterologous system

To determine whether AKAP6 is sufficient to recruit centrosomal proteins, we ectopically expressed AKAP6 in a heterologous system. Previously it has been shown that in non-muscle cells, nesprin-1α expression is required for the targeting of AKAP6 to the nuclear envelope (*Pare et al., 2005*). We first tested the ability of AKAP6 to recruit the PACT domains of Pcnt and AKAP9 in two different cell types (ARPE19 epithelial cells and NIH3T3 fibroblasts). Expression of nesprin-1α alone did not efficiently recruit the PACT domain of Pcnt or AKAP9 to the nuclear envelope (*Figure 7A* and *Figure 7—figure supplement 1A,B*). In contrast, when AKAP6-GFP was co-expressed together with nesprin-1α, the PACT domains of both Pcnt and AKAP9 were efficiently localized to the nuclear envelope (*Figure 7A*). Consistent with our previous results, the SR domains SR1-3 of AKAP6 were sufficient to recruit the PACT domain of Pcnt and AKAP9. At the same time, co-expression of AKAP6-SR1, which cannot bind to nesprin-1α, resulted in the cytosolic localization of the PACT domain (*Figure 7A* and *Figure 7—figure supplement 1A,B*).

In order to exclude that AKAP6 requires nesprin-1α or other factors localized in the nuclear envelope for the recruitment of the PACT domain, we targeted the SR1 domain of AKAP6 to the plasma membrane by tagging AKAP6-SR1 with a farnesyl-tag (tdTomato-AKAP6-SR1-Farnesylated (tdTomatoSR1-F)). Expression of tdTomatoSR1-Farn, but not tdTomato-Farnesylated alone (tdTomato-F), resulted in the recruitment of the PACT domain of both Pcnt and AKAP9 to the plasma membrane of ARPE19 and NIH3T3 fibroblasts (*Figure 7B* and *Figure 7—figure supplement 1C,D*). Furthermore, co-expression of AKAP6 and nesprin-1α, but not nesprin-1α alone, resulted in the recruitment of full-length FLAG-PcntS-Myc to the nuclear envelope of ARPE19 cells (*Figure 7C*).

## AKAP6 is sufficient to recruit endogenous centrosomal proteins to the nuclear envelope in nesprin-1α expressing epithelial cells

We then assessed whether the expression of the nesprin-1α/AKAP6 complex in a heterologous system could also induce recruitment of endogenous centrosomal proteins to the nuclear envelope. While no recruitment of endogenous Pcnt or AKAP9 was observed in ARPE19 epithelial cells expressing only nesprin-1α (*Figure 8A,B*), co-expression of AKAP6 for 30 hr induced recruitment of endogenous Pcnt and AKAP9 to the nuclear envelope in 47 ± 10% and 51 ± 13% of double transfected cells, respectively. In the majority of these cells, the nuclear envelope region proximal to the centrosome exhibited a more prominent Pcnt or AKAP9 signal (*Figure 8A,B*, stars), similar to that described during early differentiation of myocytes (*Zaal et al., 2011*). In addition, some cells were found with Pcnt or AKAP9 staining all around the nuclear envelope (*Figure 8—figure supplement 1A,B*). The percentage of cells showing endogenous recruitment of Pcnt or AKAP9 as well as the number of cells showing Pcnt and AKAP9 localization around the entire nuclear envelope of nesprin-

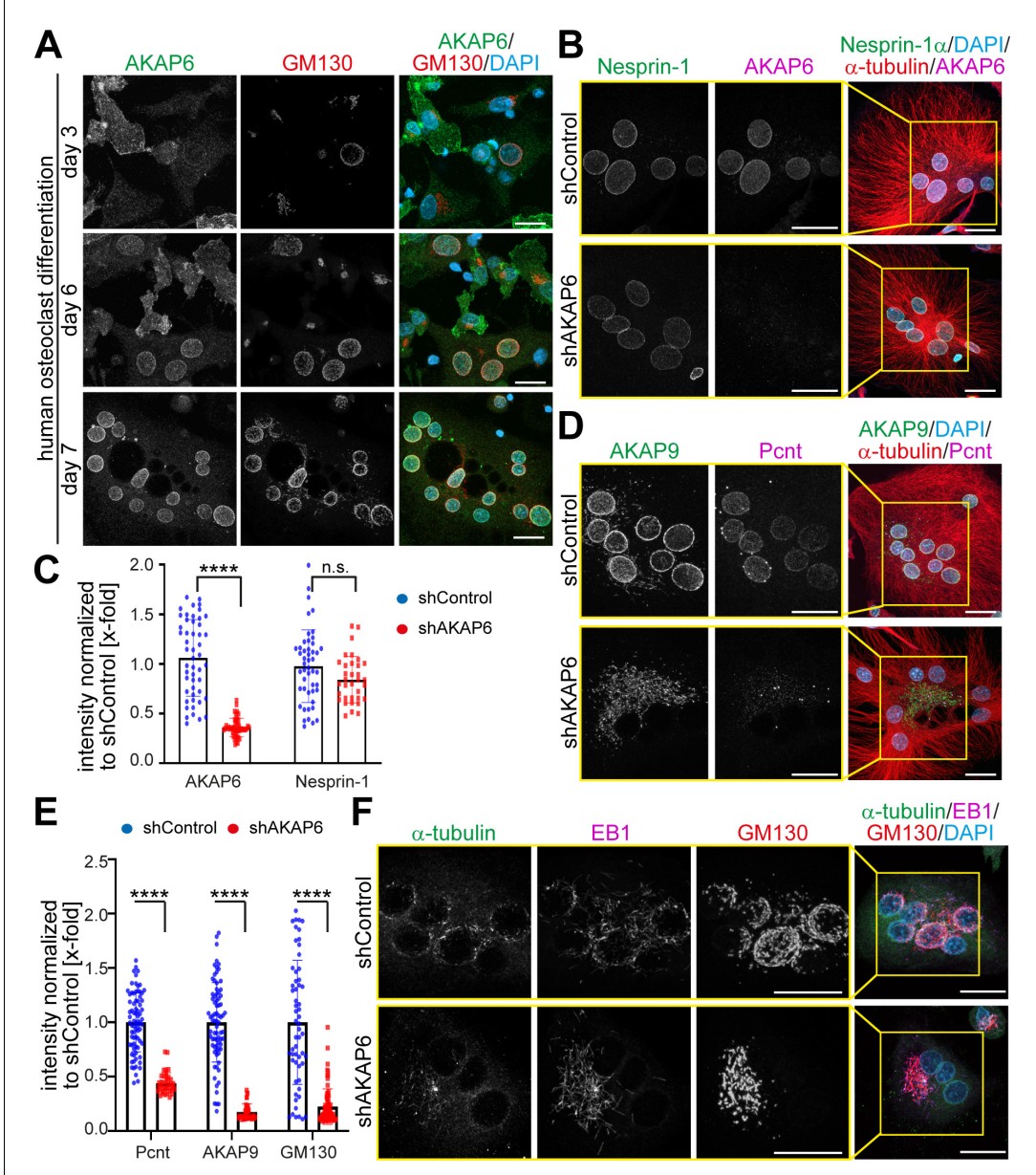

**Figure 6.** AKAP6 is a key component of nuclear envelope MTOCs. (**A**) Immunostaining of AKAP6 (green) and GM130 (red) in human osteoclasts after 3, 6 or 7 days of differentiation. Scale bars: 10 μm. (**B**) Immunostaining of nesprin-1 (green), AKAP6 (magenta), α-tubulin (red), and DNA (DAPI) in control or AKAP6-depleted osteoclasts. Scale bars: 10 μm. (**C**) Quantification of B as nuclear intensity normalized to shControl. Error bars represent the SD. n = 48, 48, 34, 34 (from left to right). Data are pooled from three independent donors. Statistical test: two-way ANOVA with post-hoc Bonferroni comparison. ****: p<0.0001, n.s.: no significance. (**D–E**) Immunostaining of AKAP9 (green), Pcnt (magenta), α-tubulin (red), and DNA (DAPI) in control and AKAP6-depleted osteoclasts. Scale bars: 10 μm. (**E**) Quantification of nuclear intensity normalized to shControl. Error bars represent the SD. n = 77, 39, 77, 39, 58, 76 (from left to right). Data were pooled from three independent donors. Statistical test: two-way ANOVA with post-hoc Bonferroni comparison. ****p<0.0001. (**F**) Immunostaining of α-tubulin (green), EB1 (magenta), GM130 (red), and DNA (DAPI) in control and AKAP6-depleted osteoclasts, after 15 s of recovery from cold-induced microtubule depolymerization. Scale bars: 20 μm.

The online version of this article includes the following source data and figure supplement(s) for figure 6:

**Source data 1.** Underlying data for graphs in panels C and E.

**Figure supplement 1.** AKAP6 is a key component of nuclear envelope MTOCs.

1α/AKAP6-positive ARPE19 cells increased with time (***Figure 8A–D*** and ***Figure 8—figure supplement 1A,B***). After 72 hr, most of the nesprin-1α/AKAP6-positive cells showed recruitment of Pcnt

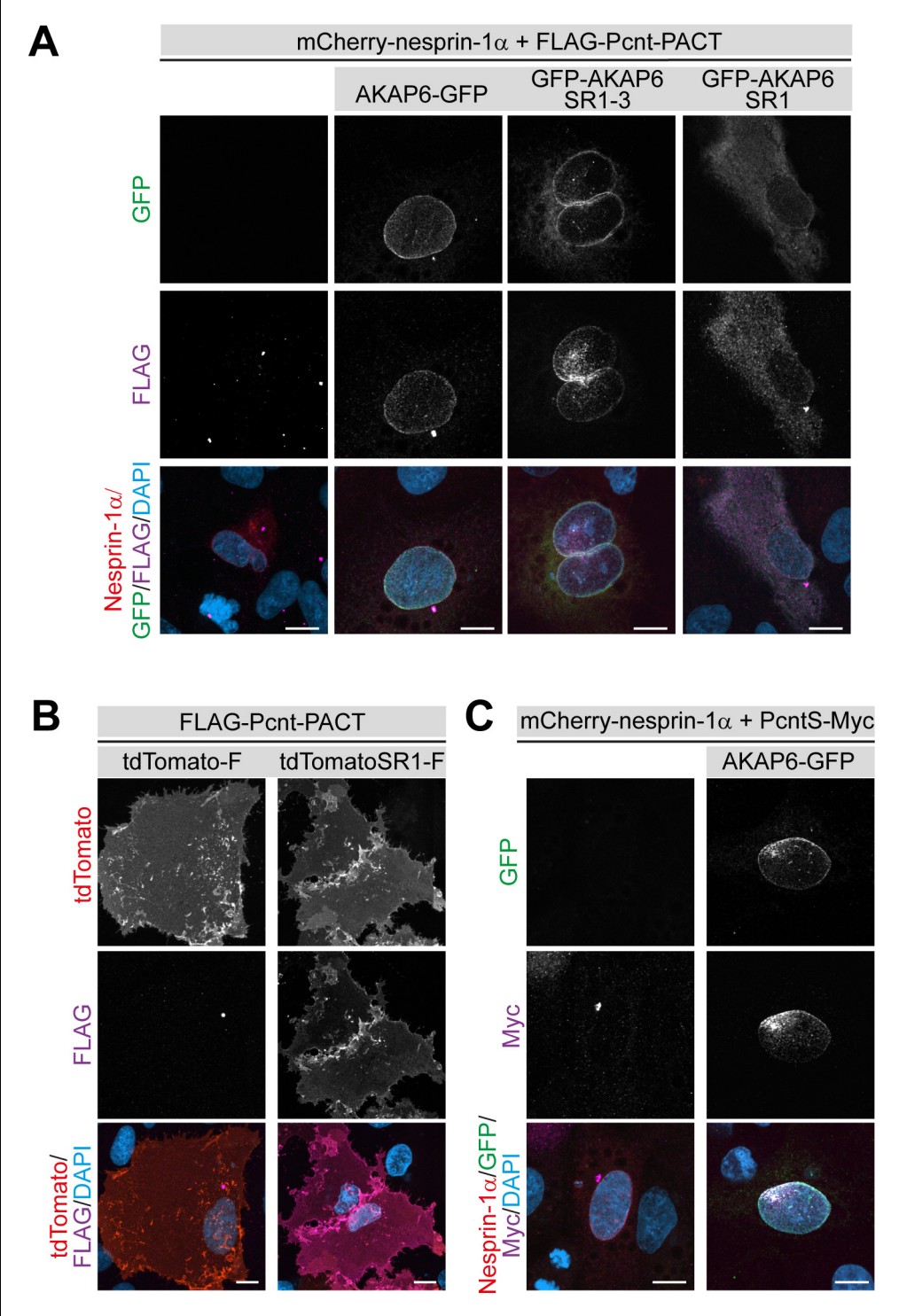

**Figure 7.** AKAP6 is sufficient to anchor the PACT domain of Pcnt and AKAP9 to the nuclear envelope of nesprin-1α expressing epithelial cells. (**A**) ARPE19 epithelial cells co-transfected with mCherry-nesprin-1α, FLAG-Pcnt-PACT, and the corresponding GFP construct, were immunostained with anti-FLAG (magenta) and DNA (DAPI). (**B**) Immunostaining of FLAG (magenta) in ARPE19 cells co-transfected with tdTomato-Farnesyl (tdTomato-F) or tdTomato-AKAP6-SR1-Farnesyl (tdTomatoSR1-F) together with FLAG-Pcnt-PACT indicating that farnesylated SR1 is sufficient to recruit PACT to the plasma membrane. (**C**) ARPE19 cells co-transfected with mCherry-nesprin-1α, FLAG-PcntS-Myc with or without AKAP6-GFP, were immunostained with anti-Myc (magenta), and DNA (DAPI). Scale bars: 10 μm.

*Figure 7 continued on next page*

*Figure 7 continued*

The online version of this article includes the following figure supplement(s) for figure 7:

**Figure supplement 1.** AKAP6 is sufficient to anchor Pcnt and AKAP9 to the nuclear envelope of nesprin-1α cells.

---

(90 ± 5%) and AKAP9 (72 ± 7%), and around 30% of the cells (29 ± 6% for Pcnt and 19 ± 9% for AKAP9) showed coverage of the entire nuclear envelope. We could not find any cell showing recruitment of Pcnt or AKAP9 in cells expressing nesprin-1α alone. Recruitment of the Golgi was less efficient, only 32 ± 8% of the nesprin-1α/AKAP6-positive cells showed a partial localization of the Golgi around the nucleus at 48 hr (*Figure 8E,F* and *Figure 8—figure supplement 1C*). At 72 hr, the number of cells with partial localization increased to 55 ± 13%, 15 ± 8% exhibited a clear crescent pattern, and in 7 ± 6% the Golgi covered the entire nuclear envelope (*Figure 8E,F*). Recruitment of the Golgi to the nuclear envelope was not found in nesprin-1α expressing cells alone. These data indicate that expression of the nesprin-1α/AKAP6 complex is sufficient to recruit centrosomal proteins and the Golgi to the nuclear envelope.

## AKAP6 is required for cell-specific functions

Previously, it has been shown that the Golgi and/or microtubules play an important role in cardiomyocyte cell growth (hypertrophy) (*Nash et al., 2019*) and bone resorption activity of osteoclast (*Guimbal et al., 2019*; *Ng et al., 2013*; *Ye et al., 2011*). Therefore, we tested whether AKAP6 is required for these cellular functions. During cardiomyocyte hypertrophy, downstream of the endothelin (ET-1A) receptor, Golgi membrane-bound phosphatidylinositol 4-phosphate (PI4P) is hydrolyzed by phospholipase C epsilon (PLCε), which is localized to the nuclear envelope via AKAP6. The hydrolysis product diacylglycerol (DAG) promotes activation of nuclear protein kinase D, which in turn, stimulates hypertrophic pathways (*Zhang et al., 2013*). This implies that a close connection between the Golgi and the nuclear envelope is crucial for proper hypertrophy activation. To test this hypothesis, we disrupted the Golgi by depleting either AKAP6 or AKAP9 and induced hypertrophy by treating P3 cardiomyocytes with endothelin-1 (ET-1). We stained cardiomyocytes for atrial natriuretic factor (ANF, also atrial natriuretic peptide, ANP), which is expressed around the nucleus in hypertrophic cardiomyocytes (*Putinski et al., 2013*). Consistent with the role of AKAP6 in recruiting PLCε to the nuclear envelope, AKAP6 depletion inhibited hypertrophy by ~50%, measured by the percentage of cardiomyocytes expressing ANF. Notably, AKAP9 depletion inhibited hypertrophy by ~80% (*Figure 9A,B*). Utilizing the PI4P reporter GFP-PH-FAPP (*Balla et al., 2005*), we confirmed that PI4P remains associated with the Golgi and loses contact to the nuclear envelope in AKAP6- and AKAP9-depleted cardiomyocytes (*Figure 9C*). Next, we determined the ability of these cells to hydrolyze PI4P. Stimulation of siControl-treated cardiomyocytes with ET-1 caused a significant depletion of perinuclear Golgi FAPP-PH-GFP fluorescence within 40 min (*Figure 9D*), consistent with its hydrolysis, as previously reported (*Zhang et al., 2013*). In contrast, depletion of AKAP6 or AKAP9 abrogated ET-1-dependent depletion of PI4P fluorescence (*Figure 9D*). These data indicate that AKAP6-AKAP9 mediated linkage of the Golgi to the nuclear envelope plays an important role in cardiac hypertrophy.

To our knowledge, no data is available in the literature indicating that a close interaction of nucleus and Golgi is required for osteoclast function. However, literature indicates that Golgi and/or microtubules are required for osteoclast function, that is bone resorption (*Ng et al., 2013*; *Ti et al., 2015*; *Ye et al., 2011*). Therefore, we tested whether AKAP6 is required for bone resorption in osteoclasts. Targeting AKAP6, either by expression of shAKAP6 (knockdown, *Figure 6*) or overexpression of its SR1 domain (displacement of Golgi and MTOC from the nuclear envelope, *Figure 6—figure supplement 1*), resulted in a reduction of ~50% in bone resorption compared to control transduced cells (*Figure 9E,F*). To ensure that the effect of AKAP6 inactivation is not due to defects in osteoclast differentiation, proper formation of multinuclear osteoclasts was confirmed upon shAKAP6 and SR1 expression by TRAP assays (*Figure 9G,H*). These data demonstrate that AKAP6 is a critical component of the ncMTOC in osteoclasts required for proper bone resorption function.

Collectively, our data indicate that AKAP6 is a key component of the nuclear envelope MTOC by linking the Golgi to the nuclear envelope, and is required for cell type-specific functions.

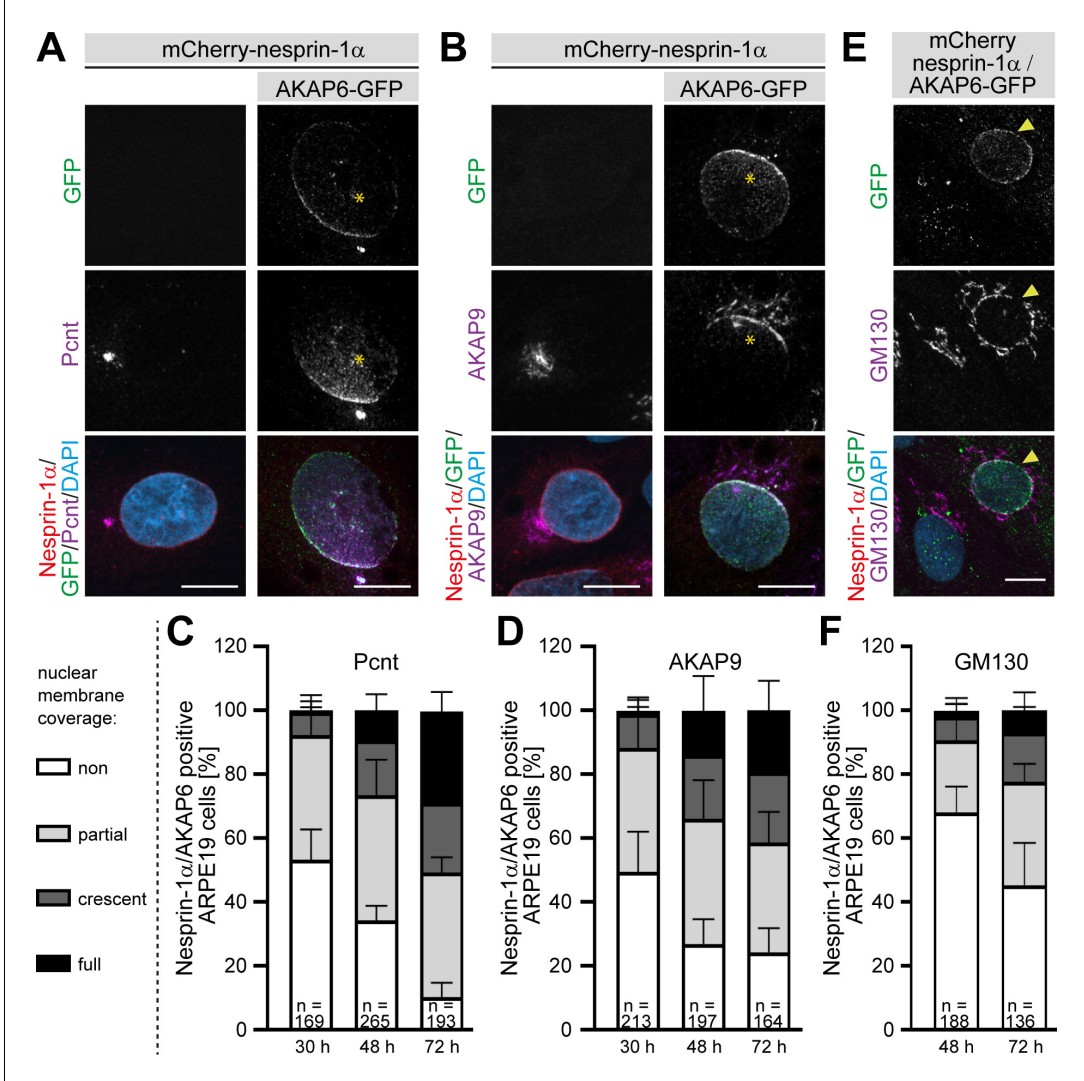

**Figure 8.** AKAP6 is sufficient to recruit endogenous Pcnt and AKAP9 to the nuclear envelope of nesprin-1α expressing epithelial cells. (A–B) ARPE19 cells co-transfected with mCherry-nesprin-1α and AKAP6-GFP and immunostained for endogenous (A) Pcnt (magenta) or (B) AKAP9 (magenta) indicating that AKAP6 can recruit endogenous centrosomal proteins to the nuclear envelope of nesprin-1α-expressing cells. Note the partial nuclear localization of Pcnt or AKAP9 near the centrosome (asterisks). (C–D) Percentage of Cherry-nesprin-1α-AKAP6-GFP expressing cells showing non, partial, crescent or full recruitment of Pcnt (C) or AKAP9 (D) to the nuclear envelope. Error bars represent the SD. Data are pooled from four independent experiments. Total number of Cherry-nesprin-1α/AKAP6-GFP expressing cells analyzed are indicated. (E) ARPE19 cells co-transfected with mCherry-nesprin-1α and AKAP6-GFP and immunostained for endogenous GM130 (magenta). Transfected cell is labeled with a yellow arrowhead. (F) Percentage of Cherry-nesprin-1α/AKAP6-GFP expressing cells showing non, partial, crescent or full recruitment of GM130 to the nuclear envelope. Data are represented as individual biological replicates, together with mean ± SD from three independent experiments. Total number of Cherry-nesprin-1α/AKAP6-GFP expressing cells analyzed are indicated. Scale bars: 10 μm.

The online version of this article includes the following source data and figure supplement(s) for figure 8:

**Source data 1.** Underlying data for graphs in panels C, D and F.

**Figure supplement 1.** AKAP6 is sufficient to induce recruitment of Pcnt, AKAP9 and GM130 to the nuclear envelope of nesprin-1α cells.

## Discussion

We conclude that AKAP6 is a key component of the nuclear envelope MTOC. By binding both nesprin-1α and Pcnt or AKAP9 through its spectrin domains, AKAP6 anchors the Golgi and centrosomal proteins at the nuclear envelope to assemble the ncMTOC. Several lines of evidence support this conclusion. First, AKAP6 is required and sufficient for centrosomal protein recruitment to the nuclear envelope. Second, AKAP6 and AKAP9 form a protein platform tethering the Golgi to the

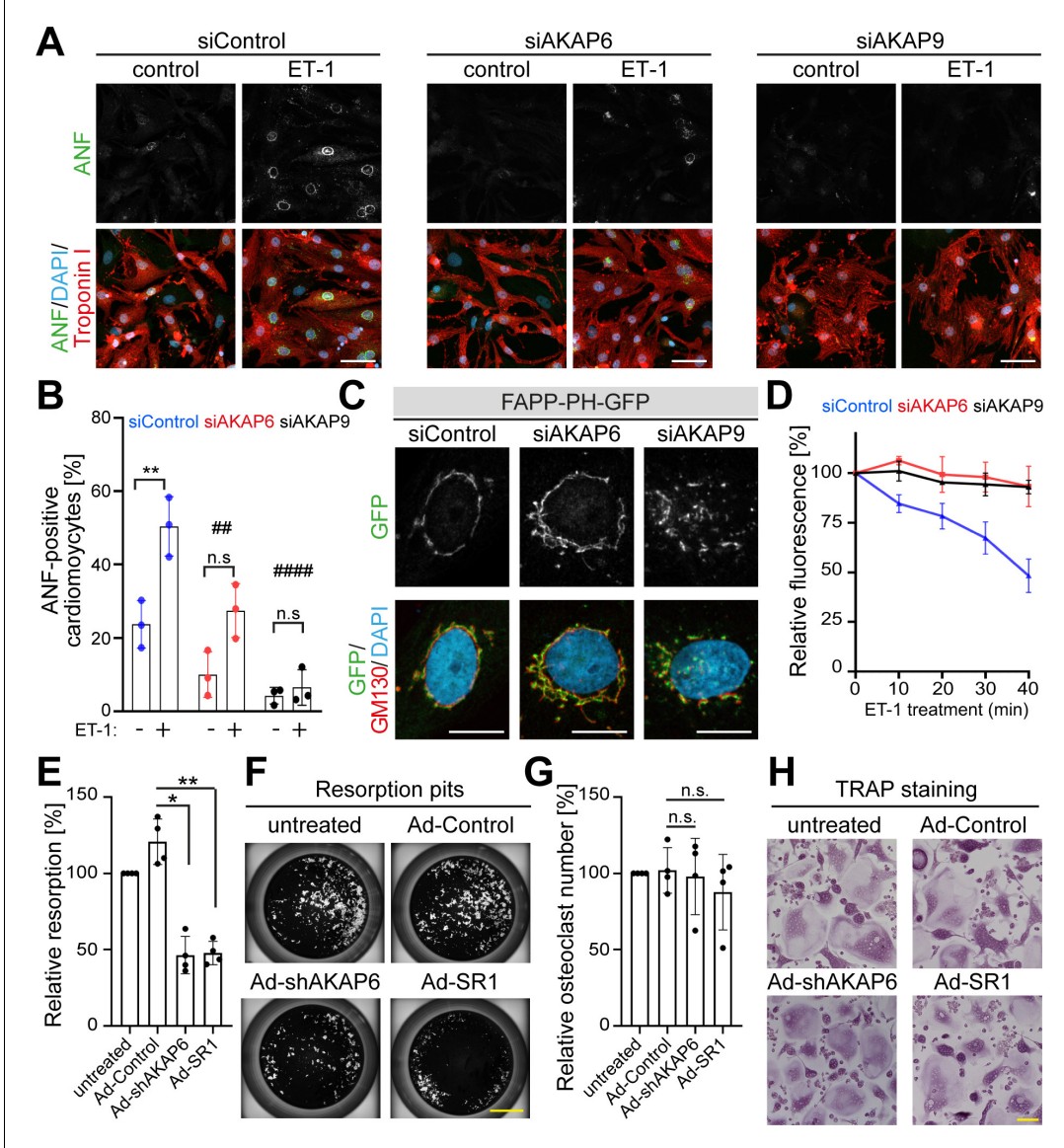

**Figure 9.** AKAP6 and AKAP9 dependent Golgi localization to the nuclear envelope is required for cell-specific functions in cardiomyocytes and osteoclasts. (**A**) Immunostaining of ANF (green), troponin I (red), and DNA (DAPI) of P3 cardiomyocytes transfected with siControl, siAKAP6, or siAKAP9 and stimulated with vehicle or 100 nM ET-1 for 24 hr. Cells were analyzed for a hypertrophic response (perinuclear ANF expression). (**B**) Quantitative analysis of the percentage of ANF-positive troponin I-positive cardiomyocytes. Data are represented as individual biological replicates, together with mean ± SD from three independent experiments. Statistical analysis was performed with two-way ANOVA with post-hoc Bonferroni comparison. *: p<0.05, **: p<0.01, ****: p<0.0001, n.s.: no significance; ##: p<0.01, ####: p<0.0001 compared to siControl + ET-1. (**C**) Immunostaining of GM130 and DNA (DAPI) in control, AKAP6- and AKAP9-depleted cardiomyocytes transfected with FAPP-PH-GFP indicating that depletion of AKAP6 and AKAP9 displaced most of the FAPP-PH-GFP signal from the nuclear envelope. Scale bars: 10 µm. (**D**) Quantitative analysis of C. Individual regions of GFP fluorescence in siControl-, AKAP6- and AKAP9-treated P3 cardiomyocytes transfected with FAPP-PH-GFP were monitored with confocal microscopy after treatment with vehicle or 100 nM ET-1 and followed for 40 min. Relative fluorescence of GFP was normalized to time 0. 15 cells per condition were measured and pooled from three independent experiments. Error bars represent the SD. (**E**) Relative resorbed area of osteoclasts transduced with the indicated adenovirus for the last 3 days of differentiation in calcium phosphate-coated wells. Data are ± SD of 4 independent experiments. Statistical analysis was performed with One-way ANOVA with Bonferroni comparison. *: p=0.015, **: p<0.01. (**F**) Representative micro images. Scale bars: 500 µm. (**G**) Relative osteoclast number after transduction of the indicated virus for the last 3 days of differentiation. TRAP-positive cells with ≥3 nuclei were considered as osteoclasts. Data are mean ± SD of 4 independent experiments. Statistical analysis was performed with one-way ANOVA with Bonferroni comparison. n.s.: no significance. (**H**) Representative images. Scale bars: 100 µm.

The online version of this article includes the following source data for figure 9:

**Source data 1.** Underlying data for graphs in panels B, D, E and G.

nuclear envelope. Around the nucleus, both membrane entities, Golgi and nuclear envelope, exhibit MTOC activity utilizing different microtubule-regulating factors. While at the Golgi, microtubule nucleation is dependent on AKAP9, possibly with the help of ninein, at the nuclear envelope both γ-TuRC binding proteins AKAP9 and Pcnt are required. Importantly, AKAP6 is not only required for cardiomyocyte ncMTOC formation, but it has a general role in mammalian ncMTOC establishment at the nuclear envelope.

AKAP6 is known to scaffold multiple signaling proteins to the nuclear envelope associated with skeletal myoblast differentiation (*Vargas et al., 2012*) and cardiac gene expression and hypertrophy (*Passariello et al., 2015*; *Kritzer et al., 2014*). Here, we show a new role of AKAP6 as a site-specific adaptor for ncMTOC formation and tethering of Golgi through the binding of AKAP9 and Pcnt. Furthermore, we show that AKAP6 expression is not only required for the establishment for a nuclear envelope MTOC in striated muscle but also in osteoclasts.

In *Drosophila* muscle and fat cells exhibit a perinuclear MTOC. Although both cell types are distinct in their molecular architecture and the mechanisms by which microtubule assembly is regulated, both require the nesprin-shot complex (*Wang et al., 2015*; *Zheng et al., 2020*). Interestingly, AKAP6 has no homologue in Drosophila. While Msp300, the nesprin-1 homologue, seems to be the primary organizer of nuclear envelope MTOC (*Zheng et al., 2020*), in mammals, nesprin-1α was not sufficient to recruit centrosomal proteins and requires AKAP6 as a bridge between nesprin-1α and centrosomal proteins.

Ectopic expression of nesprin-1α and AKAP6 in a heterologous system was sufficient to recruit endogenous Pcnt and AKAP9 to the nuclear envelope. While this recruitment was not very efficient after 30 hr of expression, it increased with prolonged expression (48 hr and 72 hr) of nesprin-1α-AKAP6. This gradual recruitment mimics what occurs during early differentiation of myoblasts (*Zaal et al., 2011*) and suggests that AKAP6 is not only required as a physical platform for anchoring proteins, but expression of the complex nesprin-1α-AKAP6 can induce the recruitment process. This induction might include signaling events as well as changes on transcriptional level considering AKAP6 being known as signaling platform (*Passariello et al., 2015*).

Previously, it has been shown that a spermatid-specific isoform of centrosomin, the *Drosophila* orthologue of CDK5RAP2, can induce MTOC formation at mitochondria (*Chen et al., 2017*) and that the tracheal cell specification factor trachealess and its target Piopio play an important role in ncMTOC formation in tracheal cells (*Brodu et al., 2010*). Our data further support that site-specific anchors exist to establish cell-type-specific MTOCs. Both nesprin-1α, as the receptor at the nuclear envelope, and AKAP6, as the platform to anchor centrosomal proteins to the nuclear envelope, are specifically expressed in muscle cells and osteoclasts to ensure proper MTOC activity at the nuclear envelope.

The Golgi itself has a well-known role in nucleating microtubules (*Mukherjee et al., 2020*; *Oddoux et al., 2013*; *Ori-McKenney et al., 2012*; *Sanders and Kaverina, 2015*; *Wu et al., 2016*; *Yalgin et al., 2015*; *Zhou et al., 2015*; *Zhu et al., 2015*). Even though there is a strong correlation in the circumnuclear localization of Golgi and MTOC at the nuclear envelope in striated muscle and osteoclasts, the contribution of the Golgi to the MTOC at the nuclear envelope has been so far underestimated. Here, we show that both, Golgi and nuclear envelope, are interconnected by the AKAP6-AKAP9 complex, with AKAP9 being the main regulator for Golgi-dependent microtubule nucleation. The fact that depletion of AKAP9 causes the disruption of the Golgi from the nuclear envelope but does not abolish microtubule nucleation implies that, at least in cardiomyocytes, a pool of microtubules is nucleated in an AKAP9-independent way at the nuclear membrane. Double depletion experiments revealed that both AKAP9 and Pcnt contribute to the microtubule nucleation at the nuclear envelope. However, since both proteins are paralogues known to compensate for each other for the binding of γ-tubulin (*Takahashi et al., 2002*), another plausible explanation would be that Pcnt compensates for the lack of AKAP9 at the nuclear envelope. The existence of two pools of microtubules is favored by the fact that, in muscle fibers, microtubules are nucleated at Golgi elements positioned at the vertices of the microtubule lattice as well as the nuclear envelope (*Oddoux et al., 2013*). However, the observations that loss of nesprin-1α in muscle cells resulted in the redistribution of the microtubule nucleation to the Golgi and that microtubule nucleation is only dependent on AKAP9 (*Gimpel et al., 2017*), raise the question whether nucleation in skeletal muscle cells occurs at the nuclear envelope or only at the Golgi. In osteoclasts, it has not yet been assessed in detail which microtubule-nucleation machinery is utilized. Overall, we hypothesize that the specific

composition of the microtubule nucleators is adapted to cell-specific functions, even though the basic mechanisms are shared among different perinuclear MTOCs.

In regards to cell-specific functions, we show here that the AKAP6-AKAP9-dependent proximity of the Golgi to the nuclear envelope is required for proper induction of hypertrophy. Activation of PLCε in cardiomyocytes induces hydrolysis of phosphatidylinositol-4-phosphate (PI4P) at the Golgi apparatus leading to release of diacylglycerol (DAG) and inactive inositol biphosphate (IP$_2$) in the vicinity of the nucleus to facilitate nuclear protein kinase D (PKD) and subsequent downstream hypertrophic pathways (*Zhang et al., 2013*). Interestingly, PLCε is localized to the nuclear envelope through interaction with the SR1 domain of AKAP6, the same domain required for AKAP9 interaction. While the role of AKAP6 in hypertrophy was attributed to the release of PLCε from the nuclear envelope (*Zhang et al., 2011*), here we propose another level of regulation by controlling the proximity of the Golgi to the nuclear envelope.

Understanding the role of the MTOC at the nuclear envelope and the possibility to modulate the activity of ncMTOCs is of potential therapeutic value. A recent study showed that perinuclear microtubules exert compressive forces on cardiomyocytes nuclei that are counteracted by desmin (*Heffler et al., 2020*). In the absence of desmin, microtubules form dynamic protrusions and drive nuclear involution that causes DNA damage, loss of association between nuclear lamina and chromatin, and ultimately substantial transcriptomic changes. Thus, targeting AKAP6 to disrupt MTOC at the nuclear envelope might be beneficial to reduce nuclear involution and decrease DNA damage in desminopathies. Moreover, it has been shown that genetic deletion of AKAP6 prevents the development of heart failure associated with long-term transverse aortic constriction, conferring a survival benefit (*Kritzer et al., 2014*). Besides, inhibiting the bone resorption activity of osteoclasts is beneficial in diseases such as osteoporosis (*Figliomeni et al., 2018*) or rheumatoid arthritis (*Schett and Gravallese, 2012*).

In summary, our study provides a deeper understanding of the establishment of ncMTOCs in differentiated cells as well as a toolbox in order to elucidate in future studies the role of MTOC at the nuclear envelope, especially in cardiomyocytes, and to harness the therapeutic potential of AKAP6 manipulation.

# Materials and methods

**Key resources table**

| Reagent type (species) or resource | Designation | Source or reference | Identifiers | Additional information |
|---|---|---|---|---|
| Strain (*Saccharomyces cerevisiae*) | AH109 | Clontech | | Reporters: *HIS3, ADE2, MEL1, LacZ* |
| Cell line (*Homo sapiens*) | HEK293 | ATCC | Cat# PTA-4488, RRID:CVCL_0045 | |
| Cell line (*H. sapiens*) | ARPE19 | ATCC | Cat# CRL-2302, RRID:CVCL_0145 | |
| Cell line (*Mus musculus*) | NIH3T3 fibroblast cell line | ATCC | Cat# CRL-1658, RRID:CVCL_0594 | |
| Transfected construct | pEGFP-N1 | Clontech | Cat# 6085–1 | |
| Transfected construct (*Rattus norvegicus*) | pEGFP-N1 AKAP6β | This paper | N/A | |
| Transfected construct (*Rattus norvegicus*) | pEGFP-N1 AKAP6βsiRNA res | This paper | N/A | |
| Transfected construct (*Rattus norvegicus*) | pEGFP -C2 AKAP6 -SR1-3 (585–1286) | This paper | N/A | |
| Transfected construct (*Rattus norvegicus*) | pEGFP -C2 AKAP6 -SR1-2 (585–1065) | This paper | N/A | |
| Transfected construct (*Rattus norvegicus*) | pEGFP -C2 AKAP6 -SR1 (585–915) | This paper | N/A | |

*Continued on next page*

Continued

| Reagent type (species) or resource | Designation | Source or reference | Identifiers | Additional information |
|---|---|---|---|---|
| Transfected construct (*Rattus norvegicus*) | pEGFP -C2 AKAP6 -SR2 (915–1065) | This paper | N/A | |
| Transfected construct (*Rattus norvegicus*) | pEGFP -C2 AKAP6 -SR3 (1065–1286) | This paper | N/A | |
| Transfected construct (*Mus musculus*) | pmCherry-Nesprin1α | This paper | N/A | |
| Transfected construct (*H. sapiens*) | pCMV-Tag2b-hAKAP9-CTermin | This paper | N/A | |
| Transfected construct (*H. sapiens*) | pCMV-Tag2b-hPCNT-CTermin | This paper | N/A | |
| Transfected construct (*H. sapiens*) | pDsRed-PACT-Myc | *Mikule et al., 2007* | | |
| Transfected construct (*H. sapiens*) | GFP-FLAG-PcntB-Myc | *Lee and Rhee, 2012* | | |
| Transfected construct | pGBKT7 | Clontech | Cat# 630443 | |
| Transfected construct (*Rattus norvegicus*) | pGBKT7-AKAP6-SR1(585–915) | This paper | N/A | |
| Transfected construct | pGADT7 | Clontech | Cat# 630442 | |
| Transfected construct (*H. sapiens*) | pGADT7-Pcnt-PACT | This paper | N/A | |
| Transfected construct (*H. sapiens*) | pGADT7-AKAP9-PACT | This paper | N/A | |
| Transfected construct | pmCherry-Farnesyl5 | Addgene | Cat# 55045 | |
| Transfected construct | tdTomato-Farnesyl5 | Addgene | Cat# 58092 | |
| Transfected construct (*Rattus norvegicus*) | tdTomato-SR1-Farnesyl5(585-915) | This paper | N/A | |
| Transfected construct (*Rattus norvegicus*) | pmCherry-Farnesyl5 -SR1(585–915) | This paper | N/A | |
| Transfected construct (*Rattus norvegicus*) | pENTR1-SR1(585–915) | This paper | N/A | |
| Transfected construct (*H. sapiens*) | pEGFP-C2-AKAP9-1KB | This paper | N/A | |
| Transfected construct | FAPP-PH-GFP | *Balla et al., 2005* | | |
| Biological sample (*Rattus norvegicus*) | E15 cardiomyocytes | This paper | N/A | |
| Biological sample (*Rattus norvegicus*) | P3 cardiomyocytes | This paper | N/A | |
| Biological sample (*H. sapiens*) | Osteoclasts | This paper | N/A | |
| Antibody | anti-PCM1 (H-262) (rabbit polyclonal) | Santa Cruz Biotechnology | Cat# sc-67204, RRID:AB_2139591 | IF: (1:500) WB: (1:1000) |
| Antibody | anti-PCM1 (G-6) (mouse monoclonal) | Santa Cruz Biotechnology | Cat# sc-398365, RRID:AB_2827155 | IF: (1:500) |
| Antibody | anti-AKAP6 (rabbit polyclonal) | Sigma-Aldrich | Cat# HPA048741, RRID:AB_2680506 | IF: (1:500) WB: (1:1000) |
| Antibody | anti-AKAP6 (mouse monoclonal) | Covance Research Products Inc | Cat# MMS-448P-100, RRID:AB_10094719 | IF: (1:500) |
| Antibody | anti-AKAP6 (rabbit polyclonal) | M.S. Kapiloff (*Li et al., 2013*) | | IF: (1:1000) |
| Antibody | anti-Pericentrin (rabbit polyclonal) | BioLegend | Cat# PRB-432C, RRID:AB_291635 | IF: (1:500) |

*Continued*

| Reagent type (species) or resource | Designation | Source or reference | Identifiers | Additional information |
|---|---|---|---|---|
| Antibody | anti-Pericentrin (mouse monoclonal) | Abcam | Cat# ab220784 | IF: (1:500) |
| Antibody | anti-nesprin-1 (MANNES1E) (mouse monoclonal) | G. Morris (*Randles et al., 2010*) | N/A | IF: (1:100) |
| Antibody | anti-nesprin-1 (rabbit polyclonal) | Covance | Cat# PRB-439P-100, RRID:AB_10094891 | IF: (1:500) |
| Antibody | anti-α-tubulin (mouse monoclonal) | Sigma-Aldrich | Cat# T9026, RRID:AB_477593 | WB:(1:1000) |
| Antibody | anti-tubulin (rat monoclonal) | Abcam | Cat# ab6160, RRID:AB_305328 | IF: (1:500) |
| Antibody | anti-Troponin I (goat polyclonal) | Abcam | Cat# ab56357, RRID:AB_880622 | IF: (1:500) |
| Antibody | anti-γ-tubulin (mouse monoclonal) | Santa Cruz Biotechnology | Cat# sc-51715, RRID:AB_630410 | IF: (1:100) |
| Antibody | anti-ninein (mouse monoclonal) | Santa Cruz Biotechnolgy | Cat# sc-376420, RRID:AB_11151570 | IF: (1:100) |
| Antibody | anti-AKAP9 (rabbit polyclonal) | Sigma-Aldrich | Cat# HPA026109, RRID:AB_1844688 | IF: (1:500) |
| Antibody | anti-AKAP9 (mouse monoclonal) | BD Biosciences | Cat# 611518, RRID:AB_398978 | IF: (1:250) |
| Antibody | anti-GM130 (mouse monoclonal) | BD Biosciences | Cat# 610823, RRID:AB_398142 | IF: (1:500) |
| Antibody | anti-Atrial Natriuretic Peptide (ANF) (rabbit polyclonal) | Millipore | Cat# AB5490, RRID:AB_2155601 | IF: (1:500) |
| Antibody | anti-GFP (mouse monoclonal) | Roche | Cat# 11814460001, RRID:AB_390913 | IF: (1:1000) WB: (1:1000) |
| Antibody | anti-GFP (rabbit polyclonal) | Novus | Cat# NB600-308, RRID:AB_10003058 | IF: (1:1000) WB: (1:1000) IP: (1.5 µl) |
| Antibody | anti-FLAG (mouse monoclonal) | Sigma-Aldrich | Cat# F1804, RRID:AB_262044 | IF: (1:2000) WB: (1:1000) IP: (1.5 µl) |
| Antibody | anti-Myc (mouse monoclonal) | Cell Signaling Technology | Cat# 2276, RRID:AB_331783 | IF: (1:1000) |
| Antibody | anti-Rabbit secondary antibody, Alexa Fluor 488 (donkey) | Thermo Fisher Scientific | Cat# A-21206, RRID:AB_2535792 | IF: (1:250) |
| Antibody | anti-Rabbit secondary antibody, Alexa Fluor 594 (donkey) | Thermo Fisher Scientific | Cat# A21207, RRID:AB_141637 | IF: (1:250) |
| Antibody | anti-Rabbit secondary antibody, Alexa Fluor 647 (donkey) | Thermo Fisher Scientific | Cat# A31573, RRID:AB_2536183 | IF: (1:250) |
| Antibody | anti-Mouse secondary antibody, Alexa Fluor 488 (donkey) | Thermo Fisher Scientific | Cat# A21202, RRID:AB_141607 | IF: (1:250) |
| Antibody | anti-Mouse secondary antibody, Alexa Fluor 594 (donkey) | Thermo Fisher Scientific | Cat# A21203, RRID:AB_2535789 | IF: (1:250) |
| Antibody | anti-Mouse secondary antibody, Alexa Fluor 647 (donkey) | Thermo Fisher Scientific | Cat# A31571, RRID:AB_162542 | IF: (1:250) |
| Antibody | anti-Goat secondary antibody t, Alexa Fluor 594 (donkey) | Thermo Fisher Scientific | Cat# A32758, RRID:AB_2762828 | IF: (1:250) |

*Continued on next page*

*Continued*

| Reagent type (species) or resource | Designation | Source or reference | Identifiers | Additional information |
|---|---|---|---|---|
| Antibody | anti-Goat secondary antibody, Alexa Fluor 647 (donkey) | Thermo Fisher Scientific | Cat# A21447, RRID:AB_2535864 | IF: (1:250) |
| Antibody | anti-rat secondary antibody, Alexa Fluor 488 (donkey) | Thermo Fisher Scientific | Cat# A-21208, RRID:AB_2535794 | IF: (1:250) |
| Antibody | anti-rat secondary antibody, Alexa Fluor 594 (donkey) | Thermo Fisher Scientific | Cat# A-21209, RRID:AB_2535795 | IF: (1:250) |
| Antibody | ECL Mouse IgG, HRP-linked whole Ab (sheep) | GE Healthcare | Cat# NA931, RRID:AB_772210 | WB: (1:5000) |
| Antibody | ECL Rabbit IgG, HRP-linked whole Ab (sheep) | GE Healthcare | Cat# NA934, RRID:AB_772206 | WB: (1:5000) |
| Sequence-based reagent | Silencer Select Negative Control No. one siRNA | Thermo Fischer Scientific | Cat# 4390843 | |
| Sequence-based reagent (*Rattus norvegicus*) | siRNA targeting sequence: rat Akap6 | Thermo Fischer Scientific | Cat# 4390771 s134273 | CAAACGACCUUGAUCAAGAtt |
| Sequence-based reagent (*Rattus norvegicus*) | siRNA targeting sequence: rat Akap9 | Thermo Fischer Scientific | Cat# 4390771 s97685 | GCUUGAACAUGCGAAAGUUtt |
| Sequence-based reagent | shRNA targeting sequence: rat/mouse/human Akap6 | This paper | N/A | CGTTTGATTTGCCTCTGCAGC |
| Sequence-based reagent | Pcnt FlexiTube siRNA | Qiagen | Cat# SI01709351 | CAGGAACUCACCAGAGACGAA |
| Sequence-based reagent (*Rattus norvegicus*) | *Akap6* RT-PCR | This paper | Forward primer: | GGGTGATTTGTTTGGATTGG |
| Sequence-based reagent (*Rattus norvegicus*) | *Akap6* RT-PCR | This paper | Reverse primer: | TGTCAGAAACACTCCGCTTG |
| Sequence-based reagent (*Rattus norvegicus*) | *Gapdh* RT-PCR | This paper | Forward primer: | CAG AAG ACT GTG GAT GGC CC |
| Sequence-based reagent (*Rattus norvegicus*) | *Gapdh* RT-PCR | This paper | Reverse primer: | AGT GTA GCC CAG GAT GCC CT |
| Commercial assay or kit | Cold Fusion Cloning Kit with Competent Cells | System Biosciences | Cat# MC010B-1 | |
| Commercial assay or kit | Neonatal Heart Disassociation Kit | Miltenyi Biotec | Cat# 130-098-373 | |
| Commercial assay or kit | M-MLV Reverse Transcriptase | Sigma | Cat# M1302 | |
| Commercial assay or kit | Lipofectamine LTX | Thermo Fisher Scientific | Cat# 15338100 | |
| Commercial assay or kit | Lipofectamine RNAiMAX | Thermo Fisher Scientific | Cat# 13778150 | |
| Chemical compound, drug | Nocodazole | Sigma-Aldrich | Cat# M1404 | |
| Chemical compound, drug | Brefeldin A (BFA) | Sigma-Aldrich | Cat# B7651 | |
| Chemical compound, drug | Endothelin 1 (ET-1) | Sigma-Aldrich | Cat# E7764 | |
| Chemical compound, drug | Fibronectin | Sigma-Aldrich | Cat# F1141 | |
| Chemical compound, drug | 4',6-diamidino-2-phenylindole (DAPI) | Carl Roth | Cat# 6335.1 | |
| Chemical compound, drug | Fluoromount-G | Thermo Fisher Scientific | Cat# 00-4958-02 | |

*Continued on next page*

*Continued*

| Reagent type (species) or resource | Designation | Source or reference | Identifiers | Additional information |
|---|---|---|---|---|
| Chemical compound, drug | Dulbecco's Modified Eagle Medium (DMEM), high glucose, GlutaMAX | Thermo Fisher Scientific | Cat# 61965059 | |
| Chemical compound, drug | DMEM 199 medium | Thermo Fisher Scientific | Cat# 41150 | |
| Chemical compound, drug | Iscove's Modified Dulbecco's Medium (IMDM), GlutaMAX Supplement | Thermo Fisher Scientific | Cat# 31980 | |
| Chemical compound, drug | Fetal Bovine Serum (FBS) | biowest | Cat# S1810 | |
| Chemical compound, drug | Horse Serum | Thermo Fisher Scientific | Cat# 16050122 | |
| Chemical compound, drug | Penicillin-Streptomycin | Thermo Fisher Scientific | Cat# 15140122 | |
| Chemical compound, drug | Gentamicin | Thermo Fisher Scientific | Cat# 15750060 | |
| Chemical compound, drug | DPBS, no calcium, no magnesium | Thermo Fisher Scientific | Cat# 14190094 | |
| Chemical compound, drug | Opti-MEM I Reduced Serum Medium | Thermo Fisher Scientific | Cat# 11058021 | |
| Chemical compound, drug | YPDA Broth | Clontech | Cat# 630306 | |
| Chemical compound, drug | SD–Leu /– Trp with Agar | Clontech | Cat# 630317 | |
| Chemical compound, drug | SD–Ade /– His /– Leu /– Trp with Agar | Clontech | Cat# 630323 | |
| Software, algorithm | Fiji | http://fiji.sc | RRID:SCR_002285 | |
| Software, algorithm | ZEN blue | Carl Zeiss AG | RRID:SCR_013672 | |
| Software, algorithm | GraphPad Prism | GraphPad Prism, | RRID:SCR_002798 | |
| Other | Zeiss LSM800 confocal laser scanning microscope with Airyscan module | Carl Zeiss AG | N/A | |
| Other | Leica TCS SP8 | Leica Microsystems | N/A | |

Further information and requests for resources and reagents should be directed to and will be fulfilled by the Lead Contact, Felix B Engel (felix.engel@uk-erlangen.de).

## Rat cardiomyocyte isolation

The investigation conforms to the guidelines from Directive 2010/63/EU of the European Parliament on the protection of animals used for scientific purposes. Extraction of organs and preparation of primary cell cultures were approved by the local Animal Ethics Committee in accordance to governmental and international guidelines on animal experimentation (protocol TS-9/2016 Nephropatho). E15 embryos were obtained from pregnant Sprague Dawley rats (from Charles River Laboratories, Cologne, Germany or own bred). Rats were first injected s.c. with 0.04 mg/kg buprenorphine and were anesthetized after 30 min by isoflurane inhalation (2 ml vaporized in a 5 L beaker). After loss of standing, eyelid and pedal reflexes, animals were sacrificed by exsanguination due to heart excision upon thoracotomy and the embryos were isolated. Hearts from embryonic day 15 (E15) and P3 rats were dissected upon decapitation with operating scissors (ROBOZ [RS-6845], no anesthesia), base with atria removed, and the remaining ventricle minced. E15 ventricular cardiomyocytes were isolated with collagenase II as previously described (*Sadoshima et al., 1992*). For the isolation of P3 cardiomyocytes, rat hearts were isolated and digested utilizing the gentleMACS Dissociation kit (Miltenyi Biotech GmbH, Bergisch Gladbach, Germany) according to the manufacturer's instructions. For P3 cardiomyocyte enrichment, cells were preplated in DMEM-F12/Glutamax TM-I (Life Technologies,

Darmstadt, Germany)/10% fetal bovine serum (FBS, Biowest, Nuaille, France)/penicillin (100 U/ml)/ streptomycin (100 µg/ml) (Life Technologies). After 1.5 hr, non-attached cells, enriched in cardiomyocytes, were collected, centrifuged for 5 min at 330 x g, resuspended in cardiomyocyte medium (DMEM-F12, Glutamax TM-I containing 3 mM Na-pyruvate, 0.2% bovine serum albumin (BSA), 0.1 mM ascorbic acid, 0.5% Insulin-Transferrin-Selenium (100x, Life Technologies) and penicillin/streptomycin (100 U/mg/ml)). E15 and P3 cardiomyocytes were seeded on glass coverslips (Ø12 mm, Thermo Fisher Scientific) coated with fibronectin (25 µg/ml in PBS) or gelatin (1% in PBS) at a density of 100,000 cells/24-well in the same cardiomyocyte medium supplemented with 2% horse serum (Life Technologies) and cytosine B-D-arabinofuranoside (AraC, Sigma).

### ARPE19, HEK293T, and NIH3T3 cells

NIH3T3, HEK293, and ARPE19 cells were purchased from ATCC. Cell types were authenticated as follows: NIH3T3, morphology; HEK293, efficiency in protein production; ARPE19, cell size and morphology. Note, cell identity is not essential for this study. All cell lines were mycoplasma-free (tested every 12 months). The cells were cultured in monolayers in Dulbecco's modified Eagle's medium (DMEM, Life Technologies) supplemented with 10% FBS and antibiotics at 37°C in a 5% $CO_2$ incubator. ARPE19 cells were cultured on DMEM-F12 supplemented with 10% FBS and antibiotics.

### Generation and stimulation of human osteoclasts

Osteoclasts were differentiated as described previously (Steffen et al., 2019). Briefly, human monocytes were purified by plastic adhesion of peripheral blood mononuclear cells that had been isolated from EDTA-blood of healthy volunteers with a Ficoll gradient (Lymphoflot, BioRad). These monocytes were differentiated into osteoclasts for 7–8 days (depending on the donor) in α-Mem medium (Invitrogen) containing 10% FBS (Gibco) and 1% penicillin/streptomycin (Invitrogen) with 30 ng/ml of M-CSF, 2 ng/ml of RANKL and 1 ng/ml of TGF-β (all Peprotech). Every 3 days, the medium was changed. For immunofluorescence staining, osteoclasts were seeded on glass coverslips (Thermo Scientific). For AKAP6 depletion, at day 5 of the culture, cells were incubated in half of the usual medium volume with the indicated amounts of an adenovirus containing a shRNA against AKAP6 or a control shRNA. After 4 hr, the second half of the medium without virus was added. After additional 2 days, osteoclast differentiation was evaluated by staining for TRAP using a Leukocyte Acid Phosphatase Kit (Sigma) according to the manufacturer's instructions. TRAP- positive cells with three or more nuclei were considered as osteoclasts.

### Osteoclast resorption assay

For the resorption assay, human osteoclasts were generated and stimulated on calcium phosphate coated plates (Corning) under the same conditions as described above. The adenovirus were added at day 5 to avoid interfering with osteoclasts differentiation. Resorption was visualized with a von Kossa staining. In brief, we lysed the cells with water and incubated the wells for 30 min with 5% silver nitrate. After extensive washing, the silver stain was developed for 1 min with 5% sodium carbonate in 25% formaldehyde and unreacted silver was removed with 5% sodium thiosulfate for 5 min. For evaluation, photos covering the total well area were taken (Keyence BZ-X710 microscope) and the percentage of the resorbed area was calculated with Adobe Photoshop CC 2018.

### Reverse transcriptase PCR (RT-PCR)

RNA was isolated from different developmental stages of rat (E11 to E20, n $\geq$ 10; P5, P10, and adult, n $\geq$ 3) using TRIzol (Life Technologies). RT-PCR was performed following standard protocols. Primers used are provided in Key resources Table.

### siRNA and plasmid transfection

Cardiomyocytes were transfected with siRNA (20 or 50 nM final concentration) by addition of transfection complexes pre-formed for 20 min, containing 0.3 µl Lipofectamine RNAiMAX (Life Technologies) per pmol of siRNA in Opti-MEM medium. A list of siRNAs used in this study is contained in the Key resources Table. Cells were processed 3 days after transfection. Plasmid transfection was carried out with 500 ng DNA per well of a 24-well plate using 1 µl Lipofectamine LTX (Thermo Fisher Scientific) according to the manufacturer's instructions.

## Plasmids

Rat cDNA was obtained from rat heart RNA using M-MLV Reverse Transcriptase for first-strand cDNA synthesis with random hexamers. Human cDNA was obtained from HeLa cell RNA using the same kit. All primers used in this study are provided in Key resources Table and *Supplementary file 1*. To generate pEGFP-AKAP6β, the cDNA of AKAP6β was obtained by PCR from rat heart cDNA and fused to pEGFP-N1 using the Cold Fusion Cloning Kit. Silent mutations of the siRNA targeting site of AKAP6 were introduced by Cold Fusion Cloning. The SR domains of AKAP6 were amplified from rat heart cDNA by PCR following restriction digest with BglII and SalI and ligated to the pEGFP-C2 vector. Nesprin-1α was amplified from C2C12 cDNA by PCR following restriction digest with EcoRI and SalI and ligated to the mCherry vector. The C-terminal domain of Pcnt and AKAP9 including their PACT domain were amplified from human cDNA by PCR following restriction digest with EcoRI and SalI and ligated to the pCMV-Tag2b and pGADT7 vectors. To generate pGBKT7-AKAP6-SR1 and pmCherry-Farnesyl-AKAP6-SR1, the cDNA of the SR1 domain of AKAP6β was obtained by PCR from pEGFP-AKAP6β and fused to pGBKT7 or pmCherry-Farnesyl5 or tdTomato-Farnesyl5 using the Cold Fusion Cloning Kit. To generate pEGFP-AKAP9-1KB, the NH-terminal fragment of AKAP9 was amplified from human cDNA by PCR following restriction digest with BglII and SalI and ligated to the pEGFP-C2 vector. The muscle-specific spliced isoform of Pcnt (PcntS) was generated by cold fusion using as template the GFP-FLAG-PcntB-Myc.

## Adenovirus production

The SR1 domain of AKAP6 (585–915) was amplified by PCR, introduced into the pENTR1A plasmid with BamHI and SalI and transferred by Gateway recombinase into the adenoviral expression plasmid pAdCMV/V5/DEST (Invitrogen). The selection of a target sequence for AKAP6 knockdown and shRNA design was done by BLOCK-iT RNAi Designer web resource. One sequence sharing 100% identity among rat, human and mouse, was chosen as a target site (CGTTTGATTTGCCTCTGCAGC), inserted into the to the BLOCK-iT U6 RNAi Entry Vector and transferred by Gateway recombinase into adenoviral expression plasmid pAd/BLOCK-iT-DEST Gateway Vector (Invitrogen) following the manufacturer manual. Finally, recombinant adenoviral vectors were produced and amplified in HEK 293A cells according to manufacturer's protocol (ViraPower Adenoviral Expression System; Invitrogen).

## Microtubule regrowth assay

Cardiomyocytes were treated with 10 μg/ml nocodazole (Sigma-Aldrich) for 3 hr in the corresponding differentiation medium. After three washes, cells were let to recover in fresh differentiation medium for 2 min at 37°C/5% $CO_2$. Cells were then fixed with methanol and stained as indicated below. In osteoclasts microtubules were depolymerized by cold-treatment. Briefly, cells were incubated at 4°C for 3 hr in the corresponding differentiation medium, after 3 hr medium was exchanged to pre-warmed medium and incubated at 37°C for 30 s before fixing with cold methanol.

## Immunofluorescence and microscopy

Antibodies used in this study are listed in the Key resources Table. Cells were fixed either in methanol for 3 min at −20°C or in 4% PFA for 15 min at room temperature (RT) and then permeabilized with 0.5% Triton X-100/PBS for 5 min at RT. Cells were blocked in 10% FBS and 0.1% saponin in PBS for 30 min at RT. Primary antibodies were incubated in blocking buffer for 1–2 hr at RT or at 4°C overnight. Following primary antibody incubation, cells were washed with PBS, incubated with fluorophore-conjugated secondary antibodies in blocking buffer for 1 hr at RT and washed with PBS. DNA was visualized with 0.5 μg/ml DAPI (4′,6′-diamidino-2-phenylindole) (Sigma) in 0.1% NP40/PBS. After DAPI staining cover slips were rinsed once with Millipore-filtered water and then mounted using Fluoromount-G mounting medium. Analysis, image acquisition and high resolution microscopy were done using a LSM800 confocal laser scanning microscope equipped with an Airyscan detector and the ZEISS Blue software with Airyscan image processing.

## Immunoprecipitation and western blot

Transfected HEK293 cells were washed with ice-cold PBS, extracted in ice-cold lysis buffer (25 mM Tris-HCl, pH 7.4, 150 mM NaCl, 5 mM EDTA and 1% NP40 supplemented with protease inhibitor

cocktails), centrifuged for 10 min at 16,000 x g and supernatants were collected. For immunoprecipitation, cell supernatants were incubated with 1.5 µl of the corresponding antibody and protein G-Sepharose beads (Amersham Bioscience). Immunoprecipitates bound to the column were collected, washed three times with lysis buffer and the proteins were eluted with Laemmli sample buffer. Samples were analyzed by SDS-PAGE (4–12% NuPAGE Novex Bis-Tris gels) under reducing conditions and transferred to nitrocellulose by wet transfer at 30 V and 300 mA for 1.5 hr in 1x transfer buffer (25 mM Tris-HCl, pH 7.5, 192 mM glycine, 0.1% SDS, 10% methanol). The membrane was then blocked with $1 \times$ TBS, 0.05% Tween 20% and 10% non-fat milk and incubated with the appropriate antibodies listed in Key resources Table. Enhanced chemiluminescence reagent (PerkinElmer, Waltham, MA) was used for protein detection.

## Yeast-two hybrid

The SR1 fragment of AKAP6 and the PACT domain of Pcnt and AKAP9 were introduced into pGADT7 and pGBKT7 vectors by PCR cloning. The *Saccharomyces cerevisiae* strain AH109 was transformed by the lithium acetate procedure as described in the instructions for the MATCHMAKER two-hybrid kit (Clontech). For colony growth assays, two colonies of each AH109 transformant were resuspended in water to an $OD_{600}$ of 0.1 per ml, 5 µl of this dilution was then applied as replicates to SD –Leu –Trp (DDO) and SD –Ade –Leu -Trp –His plates (QDO) and allowed to grow at 30˚C for 3–4 days. Interactions were scored based on growth. None of the plasmids used in this study was able to drive yeast-two-hybrid reporter activity on its own.

## Imaging of phosphoinositide reporter

P3 cardiomyocytes were plated on gelatin-coated coverslips, transfected with the corresponding siRNA at day one and grown in cardiomyocyte medium supplemented with 0.2% horse serum and AraC for three additional days. Depleted cells were transfected with the FAPP-PH-GFP plasmid 20–24 hr before the end of the experiment, and analyzed by live-imaging in a confocal microscope (Leica Microsystems). Individual regions of GFP fluorescence in cardiomyocytes were monitored and followed over time after treatment with vehicle or 100 nM ET-1 for 40 min.

## Hypertrophy assays

P3 cardiomyocytes were plated on gelatin-coated coverslips, transfected with the corresponding siRNA at day one and grown in cardiomyocyte medium supplemented with 0.2% horse serum and AraC for three additional days. ET-1 was added for the last 24 hr at a concentration of 100 nM. Cells were then fixed in 4% PFA for 15 min at RT, permeabilized with 0.5% Triton X-100/PBS for 5 min at RT and stained with anti-ANF and anti-cardiac Troponin I antibodies. Ten randomly chosen microscopic fields (0.1 mm$^2$) per experiment were analysed to determine the number of ANF-positive cardiomyocytes.

## Image preparation and statistic analysis

Images were prepared by using ImageJ, and Adobe Photoshop. All images were modified by adjustments of levels and contrast. ImageJ was used for quantification of the immunofluorescence signal intensity. Statistical comparisons and graph production were performed in Graph Prism using one or two-way ANOVA with the post-hoc Bonferrani. Three biological replicates were performed per experiment. Biological replicate means independent P3-cardiomyocyte isolations, isolation of osteoclasts from different donors, or cells freshly plated, treated, fixed and stained, and analyzed. To measure the immunofluorescence signal intensity at the nuclear envelope, every nucleus was selected as a ROI using the DAPI signal and the intensity of the corresponding signal was measured. To measure the distribution of immunofluorescence signal intensity of Golgi or microtubules around the nuclear envelope along the cell, we selected the nuclei from each cell as a ROI. Like this, the differences in nuclei size and/or circularity were avoided, and distances from the nuclear edge were more precise. For Golgi quantification, every nuclear ROI was increased in 0.2 µm steps (−2 to +2 µm). At each step a 0.2 µm wide band was created and the fluorescence mean intensity from each band was measured. For microtubule quantification, every nuclear ROI was increased in 0.5 µm steps (−4 to +10 µm), and the fluorescence mean intensity was measured from 0.5 µm wide bands. The fluorescence intensity values for each band were normalized to the mean intensity of the whole cell.

All experiments were repeated at least three times. For quantification of GFP-fusion protein overexpression experiments, 15 GFP-positive cells from 10 different fields of the coverslips were scored from three independent experiments. For endogenous recruitment in ARPE19 cells, only cells expressing low levels of AKAP6 and nesprin-1α were scored in order to avoid overexpressing artifacts.

## Acknowledgements

We thank Glenn E Morris for providing us with nesprin-1 antibody (MANNES1E), Tamas Balla for providing the plasmid FAPP-PH-GFP, Stephen Doxsey for the RFP–PeriCT-Myc construct and Kunsoo Rhee for the GFP-FLAG-PcntB-Myc plasmid. tdTomato-Farnesyl-5 and mCherry-Farnesyl-5 were a gift from Michael Davidson (Addgene plasmid # 58092 and # 55045). We thank Manfred Frasch, Hanh Nguyen and Rosa M Puertollano, for critical reading of the manuscript and all members of the Engel lab for critical discussions. This work was supported by the Emerging Fields Initiative Cell 'Cycle in Disease and Regeneration' (CYDER to FBE) and an ELAN Program Grant (ELAN-16-01-04-1-Vergarajauregui to SV) from the Friedrich-Alexander-Universität Erlangen-Nürnberg; by the German Research Foundation (DFG, INST 410/91–1 FUGG and EN 453/12–1 to FBE, and HA 8163/1–1 to US), the European Union (ERC Synergy grant 810316 4DnanoSCOPE to GS), and by the Research Foundation Medicine at the University Clinic Erlangen, Germany (to SV and FBE).

## Additional information

### Funding

| Funder | Grant reference number | Author |
|---|---|---|
| Deutsche Forschungsgemeinschaft | INST 410/91-1 FUGG | Felix B Engel |
| Deutsche Forschungsgemeinschaft | EN 453/12-1 | Felix B Engel |
| Deutsche Forschungsgemeinschaft | HA 8163/1-1 | Ulrike Steffen |
| Friedrich-Alexander-Universität Erlangen-Nürnberg | CYDER | Felix B Engel |
| Friedrich-Alexander-Universität Erlangen-Nürnberg | ELAN-16-01-04-1-Vergarajauregui | Silvia Vergarajauregui |
| European Union | ERC Synergy grant 810316 4DnanoSCOPE | George Schett |
| University Clinic Erlangen | Research Foundation Medicine | Silvia Vergarajauregui Felix B Engel |

The funders had no role in study design, data collection and interpretation, or the decision to submit the work for publication.

### Author contributions

Silvia Vergarajauregui, Conceptualization, Supervision, Funding acquisition, Investigation, Visualization, Methodology, Writing - original draft, Writing - review and editing; Robert Becker, Ingo Thievessen, Conceptualization, Investigation, Methodology, Writing - review and editing; Ulrike Steffen, Funding acquisition, Investigation, Methodology, Writing - review and editing; Maria Sharkova, Investigation, Writing - review and editing; Tilman Esser, Methodology; Jana Petzold, Investigation, Methodology; Florian Billing, Investigation; Michael S Kapiloff, Resources, Writing - review and editing; George Schett, Supervision, Funding acquisition; Felix B Engel, Conceptualization, Supervision, Funding acquisition, Visualization, Writing - original draft, Writing - review and editing

## Author ORCIDs

Silvia Vergarajauregui (iD) https://orcid.org/0000-0002-9247-6123
Florian Billing (iD) http://orcid.org/0000-0002-3874-9012
Michael S Kapiloff (iD) https://orcid.org/0000-0002-7005-6953
Ingo Thievessen (iD) http://orcid.org/0000-0003-3375-1073
Felix B Engel (iD) https://orcid.org/0000-0003-2605-3429

## Ethics

Animal experimentation: The investigation conforms to the guidelines from Directive 2010/63/EU of the European Parliament on the protection of animals used for scientific purposes. Extraction of organs and preparation of primary cell cultures were approved by the local Animal Ethics Committee in accordance to governmental and international guidelines on animal experimentation (protocol TS-9/2016 Nephropatho).

## Decision letter and Author response

Decision letter https://doi.org/10.7554/eLife.61669.sa1
Author response https://doi.org/10.7554/eLife.61669.sa2

# Additional files

## Supplementary files

- Supplementary file 1. Oligonucleotides used for generating constructs.
- Transparent reporting form

## Data availability

All data generated or analysed during this study are included in the manuscript and supporting files. Source data files have been provided for all graphs.

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
