## [Decision Letter]

**Acceptance summary:**

While the composition and assembly of the centrosomal MTOC is still not fully understood, even less is known about non-centrosomal MTOCs. In this study Vergarajauregui et al. identify AKAP6 as a crucial assembly factor for the MTOC at the nuclear envelope in muscle cells and osteoclasts, by recruiting centrosomal proteins. Moreover, in cooperation with AKAP9, AKAP6 tethers the Golgi to the nuclear periphery, which also functions as MTOC in these cells.

**Decision letter after peer review:**

Thank you for submitting your article "AKAP6 orchestrates the nuclear envelope microtubule organizing center by linking Golgi and nucleus via AKAP9" for consideration by *eLife*. Your article has been reviewed by three peer reviewers, including Jens Lüders as the Reviewing Editor and Reviewer #1, and the evaluation has been overseen by Suzanne Pfeffer as the Senior Editor.

The reviewers have discussed the reviews with one another and the Reviewing Editor has drafted this decision to help you prepare a revised submission.

Summary:

This study unveils the mechanisms of non-centrosomal microtubule organization from perinuclear sites in specialized vertebrate cell types, identifying AKAP6 as a key adapter molecule that recruits microtubule nucleation complexes. It is further shown that both nuclear envelope and Golgi-associated protein complexes contribute to perinuclear microtubule organizing activity.

Overall all three reviewers agreed that this is an important contribution and that the data is of high quality and well-presented. There are two major points that we would like you to address.

Essential revisions:

1) The authors performed an AKAP6 rescue experiment and checked whether Pcnt and AKAP9 could still localise to the nuclear envelope. Instead of measuring the fluorescent intensity of Pcnt and AKAP9, as they had done in Figure 1C, they quantified the percentage of cells with a "positive" nuclear signal (Figure 1—figure supplement 2F, H). This may hide a potential incomplete rescue (perhaps the signal is reduced). We ask the authors to redo this analysis as before with fluorescence intensity measurements.

2) In Figure 7, we suggest that you demonstrate that AKAP6/nesprin-1α are sufficient for the re-localization of endogenous centrosome proteins, instead of performing their assays with overexpressed forms of pericentrin. This would make the main argument of the manuscript much stronger.

---

## [Author Response]

Essential revisions:1) The authors performed an AKAP6 rescue experiment and checked whether Pcnt and AKAP9 could still localise to the nuclear envelope. Instead of measuring the fluorescent intensity of Pcnt and AKAP9, as they had done in Figure 1C, they quantified the percentage of cells with a "positive" nuclear signal (Figure 1—figure supplement 2F, H). This may hide a potential incomplete rescue (perhaps the signal is reduced). We ask the authors to redo this analysis as before with fluorescence intensity measurements.

We thank the reviewers for pointing out this issue. To address the issue, the experiments were repeated and analyzed measuring the fluorescence intensity at the nuclei as suggested. The mean intensity of both Pcnt and AKAP9 at the nuclear envelope of rescued cells were comparable to those of siControl cells. The respective graphs (Figure 1—figure supplement 2F, H) have been exchanged.

2) In Figure 7, we suggest that you demonstrate that AKAP6/nesprin-1α are sufficient for the re-localization of endogenous centrosome proteins, instead of performing their assays with overexpressed forms of pericentrin. This would make the main argument of the manuscript much stronger.

These experiments were performed and shown as part of Figure 7. We have discussed this issue with the editor. In order to emphasize these results, we have separated in the revised manuscript the data regarding the recruitment of overexpressed proteins (new Figure 7) and endogenous proteins (new Figure 8).